# A splicing isoform of TEAD4 attenuates the Hippo–YAP signalling to inhibit tumour proliferation

Yangfan Qi[1,*], Jing Yu[2,*], Wei Han[1,*], Xiaojuan Fan[3], Haili Qian[4], Huanhuan Wei[3], Yi-hsuan S. Tsai[5], Jinyao Zhao[1], Wenjing Zhang[1], Quentin Liu[1], Songshu Meng[1], Yang Wang[1] & Zefeng Wang[3]

Aberrant splicing is frequently found in cancer, yet the biological consequences of such alterations are mostly undefined. Here we report that the Hippo–YAP signalling, a key pathway that regulates cell proliferation and organ size, is under control of a splicing switch. We show that TEAD4, the transcription factor that mediates Hippo–YAP signalling, undergoes alternative splicing facilitated by the tumour suppressor RBM4, producing a truncated isoform, TEAD4-S, which lacks an N-terminal DNA-binding domain, but maintains YAP interaction domain. TEAD4-S is located in both the nucleus and cytoplasm, acting as a dominant negative isoform to YAP activity. Consistently, TEAD4-S is reduced in cancer cells, and its re-expression suppresses cancer cell proliferation and migration, inhibiting tumour growth in xenograft mouse models. Furthermore, TEAD4-S is reduced in human cancers, and patients with elevated TEAD4-S levels have improved survival. Altogether, these data reveal a splicing switch that serves to fine tune the Hippo–YAP pathway.

[1] Institute of Cancer Stem Cell, Second Affiliated Hospital Collaborative Innovation Center of Oncology, Dalian Medical University, Dalian 116044, China. [2] Department of Biostatistics, University of North Carolina at Chapel Hill, Chapel Hill, North Carolina 27599, USA. [3] Key Laboratory of Computational Biology, CAS-MPG Partner Institute for Computational Biology, Shanghai Institutes for Biological Sciences, Chinese Academy of Sciences, Shanghai 200031, China. [4] State Key Lab of Molecular Oncology, Peking Union Medical College and Chinese Academy of Medical Sciences, Beijing 100021, China. [5] Lineberger Comprehensive Cancer Center, University of North Carolina at Chapel Hill, Chapel Hill, North Carolina 27599, USA. * These authors contributed equally to this work. Correspondence and requests for materials should be addressed to Y.W. (email: yangwang@dlmedu.edu.cn) or to Z.W. (email: wangzefeng@picb.ac.cn).

Hippo–YAP signalling is a key pathway that regulates cell proliferation, cell contact inhibition and organ size[1–3]. The transcriptional output of this pathway is mediated by TEAD proteins that partner with YAP to activate genes that stimulate cell proliferation[4,5]. As a key effector of the Hippo pathway, YAP lacks DNA-binding motif, and thus recognizes its targets through interacting with TEAD proteins[6]. Under normal condition, YAP is translocated into the nucleus to promote cell growth; however, the activation of Hippo causes YAP phosphorylation, leading to cytoplasmic retention and degradation of YAP[7,8]. Thus, defects of the Hippo signalling cause overgrowth phenotypes due to deregulation of proliferation and apoptotic defects[9,10]. Hippo–YAP pathway is directly involved in cancer development[11–13], and inhibition of the YAP activity provides a valuable route for cancer prevention and treatment[14–16]. In current model, Hippo signalling is mainly regulated via protein phosphorylation and degradation[10,17]. Intriguingly, some key components of the Hippo–YAP pathway undergo extensive regulation in RNA level through alternative splicing (AS), a major mechanism to expand coding capacity of human genome. For example, MST1 has multiple isoforms with C-terminal truncations, and YAP has eight splicing isoforms with different internal sequences[18]. However, the biological functions of these isoforms are unclear.

AS elicits control over the major hallmarks of cancer[19–21], including apoptosis[22], angiogenesis[23] and epithelial–mesenchymal transition (EMT)[24]. However, the functional consequences of most cancer-related splicing alterations are undefined. AS is generally regulated by splicing factors that specifically bind cis-elements in pre-messenger RNA (mRNA) to affect the splicing efficiency[25–27]. The alteration of splicing factor levels and activities is a main cause of splicing dysregulation in cancer[28–30]. For example, a general splicing factor, RBM4, functions as a potent tumour suppressor by controlling AS events critical to cell proliferation, migration and apoptosis[31,32]. Expression of RBM4 is significantly reduced in lung cancer and strongly correlated with patient survival, making it a valuable prognostic predictor.

Here we report that the Hippo pathway is under control of a new splicing switch in TEAD4, producing a truncated isoform, TEAD4-S, which lacks an N-terminal DNA-binding domain, but maintains YAP interaction domain. TEAD4-S is located in both the nucleus and cytoplasm, acting as a dominant negative isoform to YAP activity. Splicing of TEAD4-S is facilitated by the tumour suppressor RBM4. Consistently, TEAD4-S is reduced in cancer cells, and its re-expression suppresses cancer cell proliferation and migration, inhibiting tumour growth in xenograft mouse model. Furthermore, TEAD4-S is reduced in human cancers, and patients with elevated TEAD4-S levels have improved survival. Altogether, these data reveal a splicing switch that serves to fine tune the Hippo–YAP pathway. To our knowledge, this is the first report that the Hippo–YAP pathway is regulated through RNA splicing, which probably exemplify a new general regulatory mode of cell proliferation in RNA level. We expect that splicing misregulations of other genes in the Hippo–YAP pathway also play critical roles in cancer development and thus should be explored as new route of potential cancer therapy.

## Results

### Identification of an AS isoform of TEAD4.
The transcription factor-mediating YAP-dependent gene expression, TEAD4, contains two functional domains: an N-terminal domain specifically recognizing the enhancer of targeted genes through direct binding of DNA and a C-terminal motif that interacts with transcription co-activator YAP to promote transcription of target genes[33]. On the basis of the annotation of the RNA sequencing (RNA-seq) data, we discovered that the exon 3 of human TEAD4 can be skipped to generate a new isoform, producing a truncated TEAD4 protein with an alternative start site at exon 6. This short isoform of TEAD4 (TEAD4-S) lacks an N-terminal DNA-binding domain, but still contains an intact C-terminal YAP-binding motif (Fig. 1a; Supplementary Fig. 1a,b).

To validate the presence of this TEAD4-S isoform, we measured the levels of TEAD4-S mRNA and protein in different human tissues. While the full-length TEAD4 (TEAD4-FL) represents the major form in all tested tissues, the TEAD4-S mRNA is ubiquitously spliced with small tissue-specific variations (Supplementary Fig. 1c). Consistently, using an antibody against the common sequence of both TEAD4 isoforms (amino acids 151–261), we detected TEAD4-S protein in all tissues with abundance comparable to a canonical isoform in some tissues (for example, spleen and skeleton muscle; Supplementary Fig. 1d). These two TEAD4 isoforms have distinct subcellular localizations: while canonical TEAD4-FL is primarily located in the nucleus, TEAD4-S is found in both the nucleus and cytoplasm with no obvious preference (Fig. 1b). Importantly, both TEAD4 isoforms were found to bind YAP protein in a co-immunoprecipitation assay (Fig. 1c), supporting the finding that TEAD4-S still contains an intact C-terminal YAP-binding motif.

### Splicing of TEAD4-S is regulated by RBM4.
To identify the possible splicing factors that control the AS of TEAD4, we analysed the sequence of exon 3 in TEAD4 and discovered a putative RNA motif resembling a known binding site of RBM4 (CGGCCGG)[34]. RBM4 is a general splicing factor that functions as a tumour suppressor in a number of human cancers[31]. This observation raises the possibility that RBM4 may directly control TEAD4 splicing. To test this possibility, we generated 293 cells that stably express RBM4 on tetracycline induction, and found that RBM4 promotes the splicing of TEAD4-S in a time-dependent manner (Fig. 1d; Supplementary Fig. 1e). Such regulation is consistent across various cell types, as overexpression of RBM4 increased the TEAD4-S mRNA and protein levels in various cultured cancer cells from pancreatic, lung, liver and breast cancers (Fig. 1e; Supplementary Fig. 1f).

Next, we generated a minigene reporter containing the exon 2–4 of TEAD4 (Fig. 1f, upper panel). Consistent with the previous results in Fig. 1d,e, RBM4 expression indeed caused skipping of exon 3 in this artificial reporter (Fig. 1f, lower-left panel). Remarkably, a mutant splicing reporter destroying the putative RBM4-binding site (mut 1, CTTATA) abolished the splicing regulation of TEAD4 by RBM4, whereas another mutant reporter with a replaced RBM4-binding motif (mut 2, GTAACG)[35] restored the RBM4 regulation (Fig. 1f, lower-right panel). We further confirmed that RBM4 directly bound to TEAD4 pre-mRNA using RNA-IP (Fig. 1g). Consistently, the mutant RNA with the destroyed RBM4-binding site (mut 1) failed to interact with RBM4, while the replacement of this site with a different RBM4-binding sequence (mut 2) reinstated the RNA–RBM4 interaction (Fig. 1h). Furthermore, we used an antisense oligonucleotide (ASO3, TAATGGTGCCGGCCGT GCCC) to block the RBM4-binding site in exon 3 of TEAD4. As expected, RBM4 could no longer regulate TEAD4 splicing in the presence of antisense oligos (Supplementary Fig. 1g). Collectively, these data suggest that RBM4 indeed recognizes this putative binding site to control TEAD4 splicing.

### TEAD4-S antagonizes TEAD4-FL to repress YAP signalling.
Since TEAD4-S lacks the DNA-binding activity, it may lack the ability to promote transcription and even compete with the

canonical TEAD4 with its YAP-binding domain. We directly test this hypothesis with a reporter system containing a luciferase gene driven by a TEAD-dependent promoter[36]. We found that expression of YAP alone or co-expression of YAP/TEAD4-FL significantly increased luciferase activity, whereas co-expression of TEAD4-S/YAP suppressed the gene activation in a dose-dependent manner, suggesting that TEAD4-S has an opposite activity to canonical TEAD4. Importantly, when increasing amount of TEAD4-S was co-expressed with TEAD4-FL, TEAD4-S can inhibit the transcriptional activity of TEAD4-FL in a dose-dependent manner (Fig. 2a; Supplementary Fig. 2a). We further demonstrated that TEAD4-S can directly disrupt the binding of canonical TEAD4-FL to YAP as judged by reciprocal co-immunoprecipitation assays (Fig. 2b,c; Supplementary Fig. 2b), suggesting that TEAD4-S functions as a dominant negative isoform via competing with TEAD4-FL. In addition, while

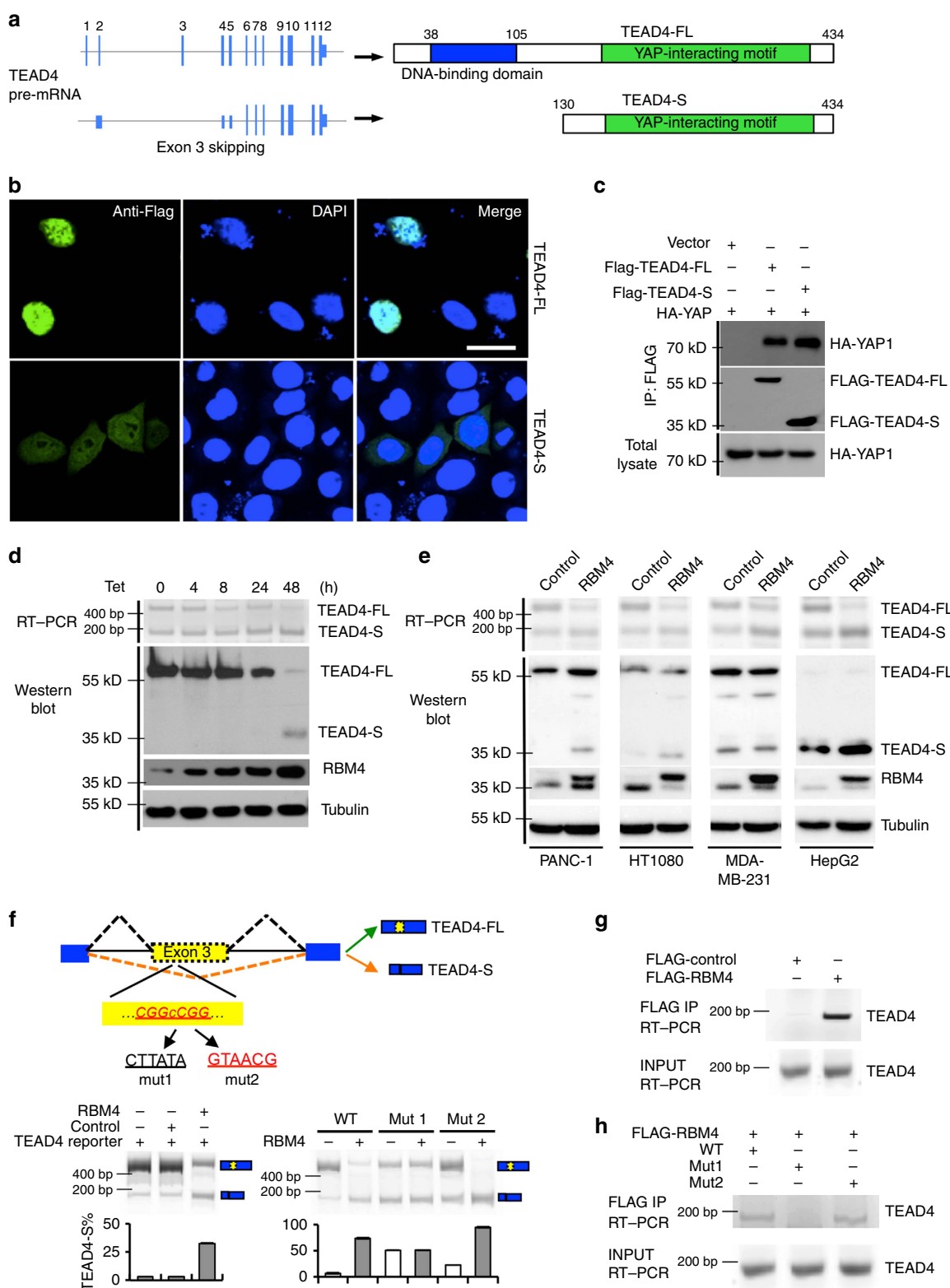

TEAD4-FL is found to bind the promoter of its endogenous target CTGF, TEAD4-S cannot bind the same DNA as judged by chromatin immunoprecipitation, presumably due to lack of the N-terminal DNA-binding domain (Fig. 2d; Supplementary Fig. 2c). Together, these data suggest that TEAD4-S functions as a dominant negative isoform.

In line with the finding that RBM4 promotes the splicing of TEAD4-S, co-expression of RBM4 with YAP also reduced TEAD4-mediated gene activation, whereas expression of RBM4 alone had no effect (Fig. 2a). This observation was further supported by measuring the endogenous targets of YAP/TEAD4, as co-expression of YAP/TEAD4-S diminished the YAP-dependent activation of CTGF and ITGB in two different cell lines (Fig. 2e,f; Supplementary Fig. 2d). Consistently, co-expression of RBM4 with YAP can also reduced the activation of these two genes by YAP (Fig. 2e,f), while expression of RBM4 alone had no effect (Supplementary Fig. 2e), further confirming that RBM4-mediated new TEAD4 splicing switch to control the YAP signalling pathway.

To further assess the impact of the RBM4-mediated TEAD4 splicing switch on YAP signalling and cellular functions, particularly cell proliferation, we conducted transcriptome profiling by RNA-seq in cells transfected with YAP alone, YAP/TEAD4-FL, YAP/TEAD4-S and YAP/RBM4 (Supplementary Fig. 2f; Supplementary Data 1). As expected, cells with co-expression of YAP with TEAD4-FL or TEAD4-S displayed limited overlapping between significantly affected genes (Supplementary Fig. 2g). Consistent with the finding that RBM4 increases splicing of TEAD-S, the genes that were up- or downregulated on YAP/TEAD4-S expression significantly overlap with those altered in the same directions by YAP/RBM4 expression, whereas the genes altered in different directions are mutually exclusive (Supplementary Fig. 2h; $P < 2.2 \times 10^{-16}$ by $\chi^2$-test).

By comparing all samples with the vector-transfected control, we identified a set of 429 YAP-activated genes that were enhanced by co-expression of TEAD4-FL, but were repressed by co-expression of TEAD4-S or RBM4 (Fig. 2g; Supplementary Data 2). Amazingly, these genes form a densely connected network as evaluated by the analysis of STRING database of gene interaction (Fig. 2h). The largest cluster of genes contains many key regulators of cell cycle, including transcription factor (for example, E2F1 and Myc), some cyclins (for example, CCNB1 and CCNE2) and genes required for chromatin alignment and segregation (for example, NDC80, CDCA8, SPDL1 and CENPF). In addition, the genes differentially regulated by two TEAD4 isoforms also include components in key signalling pathways that

affect cell proliferation (for example, CSNK2A1, PCNA and other components of the Hippo pathway), as well as genes involved in RNA splicing (for example, SMN2, SF3A2 and many splicing factors) and translation (for example, eIF3C, eIF1B and ribosomal proteins).

In addition, gene ontology analysis also suggested that the cell cycle, cell proliferation and RNA processing are main functional pathways that are activated by YAP YAP TEAD4, but diminished by the non-canonical TEAD4 isoform (Supplementary Fig. 2i). Taken together, these data indicate that the short isoform of TEAD4 harbours functions to antagonize conventional TEAD4, thus attenuating the Hippo–YAP–TEAD signalling cascade.

**TEAD4-S represses cancer cell proliferation and EMT.** Hippo–YAP pathway is known to control tumorigenesis[10]. The negative regulation of this pathway by the short isoform of TEAD4 suggests that TEAD4-S may repress cancer cell proliferation. To test this possibility, we measured the relative levels of TEAD4 in a panel of lung cancer cells. We found that, while TEAD4-S is the major isoform in the normal lung fibroblast and bronchial epithelial cells (Fig. 3a; HFL1 and HBE), all tested lung cancer cells have reduced TEAD4-S level compared with the normal cells (Fig. 3a). Furthermore, in lung cancer cells with stable overexpression of YAP alone or co-expression of YAP with TEAD4-FL, TEAD4-S or RBM4 (Supplementary Fig. 3a), TEAD4-S inhibited cell proliferation as compared with vector or TEAD4-FL (Fig. 3b,c). Such inhibition was comparable to that of RBM4 in both anchorage-dependent and -independent cell growth in two distinct cancer cell lines (Fig. 3b,c).

Hippo–YAP signalling has also been shown to regulate the EMT of cancer cells[37,38]. The inhibition of TEAD4-S to YAP-activated cell proliferation makes us to speculate that it may also repress the EMT. To test this, we examined whether the two isoforms of TEAD4 have distinct impacts on the EMT. As shown in Fig. 3d, co-expression of YAP/TEAD4-FL strongly induced EMT in two primary tumour cell lines as judged by increased levels of N-cadherin and Vimentin; however, the expression of TEAD4-S or RBM4 suppressed the induction of EMT markers (Fig. 3d). Interestingly, in H157 cells co-expressing YAP and TEAD4-FL, the majority of YAP is retained in the nucleus as pointed dots that are co-localized with TEAD4-FL (Fig. 3e; Supplementary Fig. 3b). However, in cells expressing TEAD4-S, the YAP is found in both nuclear and cytoplasmic compartments, presumably through the interaction with TEAD4-S that is diffused into the cytoplasm (Fig. 3e; Supplementary Fig. 3b).

**Figure 1 | Identification of an alternative splicing isoform of TEAD4 regulated by RBM4.** (**a**) Schematics of human TEAD4 pre-mRNA and protein. The full-length TEAD4 contains 1–12 exons, including the DNA-binding domain in the N terminal, whereas exon 3 of TEAD4 can be skipped to generate a new isoform whose translation is started at exon 6 to produce a truncated TEAD4-S protein. TEAD4-S lacks an N-terminal DNA-binding domain, but still contains an intact C-terminal YAP-binding motif. (**b**) The localizations of TEAD4-FL and TEAD4-S were determined by immunofluorescence assay. Flag-tagged TEAD4-FL or TEAD4-S was transfected into cells. The immunofluorescence assay was performed with anti-Flag antibody to show the localizations of Flag-TEAD4. Scale bar, 25 μm. (**c**) Interaction of two TEAD4 isoforms with YAP protein as judged by co-IP experiment. The Flag-tagged TEAD4-FL or TEAD-S was co-transfected with HA-tagged YAP into 293T cells, and the binding of YAP was detected. (**d**) 293 cells expressing RBM4 on tetracycline induction were collected at different time points after induction to determine the expression levels of TEAD4-FL and TEAD4-S. (**e**) RBM4 regulates the splicing of TEAD4 in various cancer cells, including PANC1, HT1080, MDA-MB-231 and HepG2. The cells were stably transfected with RBM4 or vector control, and the splicing of TEAD4 was examined by RT–PCR. The representative gels were shown. (**f**) The schematic of TEAD4 pre-mRNA where the potential RBM4-binding site (CGGCCGG) in red. TEAD4 splicing reporters with the indicated mutations (mut1: CTTATA and mut2: GTAACG) were generated (upper panel). TEAD4 splicing reporters containing wild-type RBM4-binding site (lower-left panel) or mutations (lower-right panel) were co-expressed with RBM4 or vector control in 293T cells to assay for the splicing change of TEAD4. Representative gels from triplicate experiments were shown, with the mean ± s.d. of TEAD4-S% plotted below representative gels. (**g**) Binding of TEAD4 pre-mRNAs with RBM4 was examined by RNA-immunoprecipitation assay in cells exogenously expression FLAG-RBM4 or vector control. (**h**) 293 cells were co-transfected with Flag-RBM4 or vector control and the indicated mutant or wild-type (WT) TEAD4 reporters, and then immunoprecipitated with anti-Flag antibody. The co-precipitated RNAs were detected by RT–PCR.

To further examine the functional role of TEAD4 splicing switch, we applied a recently developed approach, engineered splicing factors (ESFs)[39–42], to specifically manipulate TEAD4 splicing and to test whether the splicing changes of TEAD4 can directly affect YAP-mediated EMT and tumour proliferation. We designed and generated a PUF domain that can specifically bind

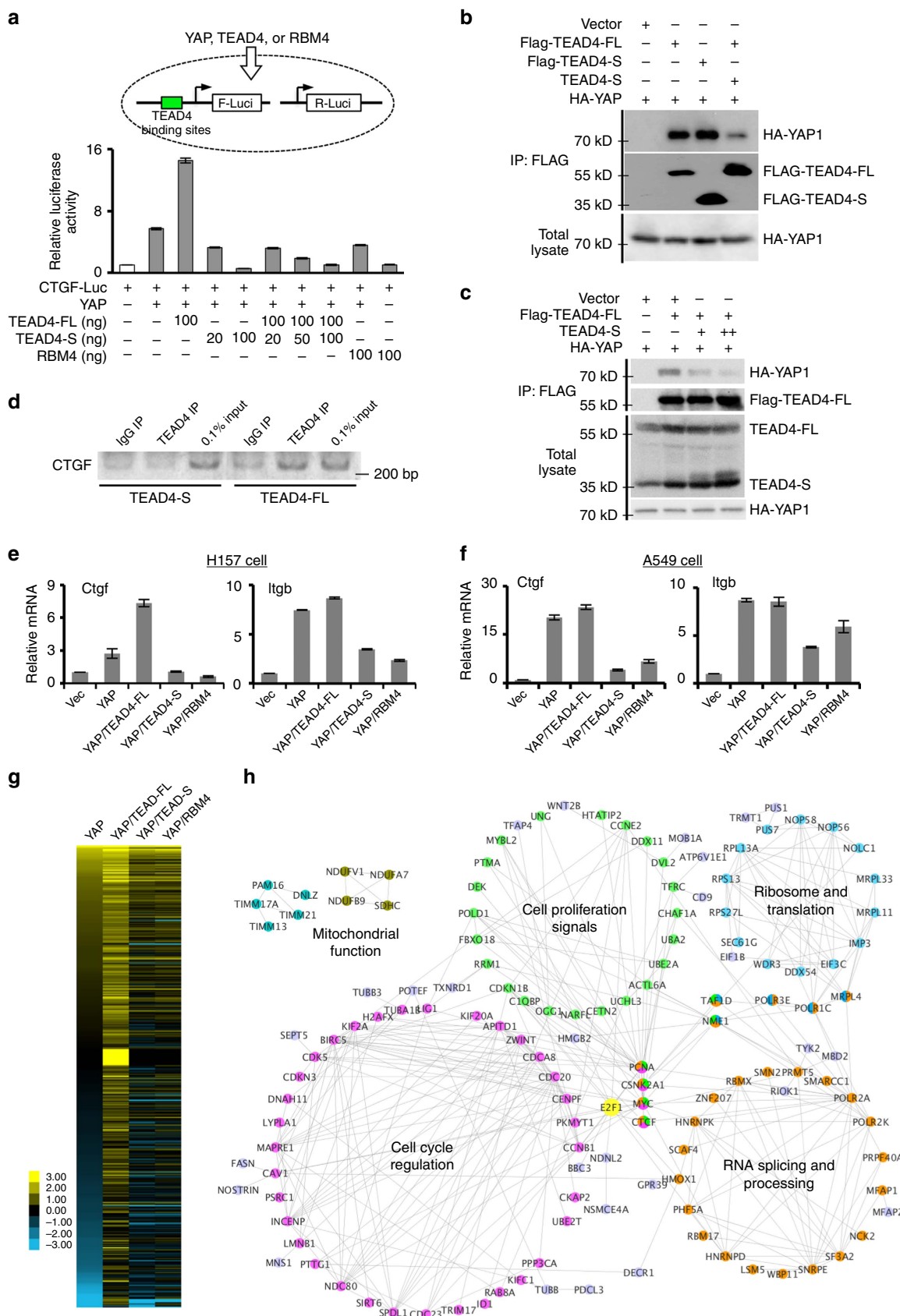

to an 8-nt RNA sequence (GAGCCTTG) in exon 3 of TEAD4, and fused the designer PUF domain with a glycine-rich motif that can inhibit exon inclusion on binding to pre-mRNA. The resulting ESF, ESF-TEAD4, is designed to bind and inhibit the inclusion of exon 3 in TEAD4, thus promoting the TEAD4-S splicing (Fig. 4a). When transfecting ESF-TEAD4 in YAP stably expressed lung cancer cells, H157 and A549, we found that the ESF-TEAD4 specifically shifted TEAD4 splicing towards the short isoform as compared with control ESF (Fig. 4b). As expected, the splicing switch to TEAD4-S can indeed suppress the YAP-mediated EMT (Fig. 4c) and the tumour cell proliferation (Fig. 4d) in both lung cancer cell lines tested.

**TEAD4-S inhibits tumour growth *in vivo*.** To further assess whether TEAD4-S affects cancer growth *in vivo*, we determined whether expression of TEAD4-S could suppress tumour growth in a xenograft mouse model. We generated H157 cells with stable expression of YAP, YAP/TEAD4-FL, YAP/TEAD4-S or YAP/RBM4, and injected these cells subcutaneously into the flanks of nude mice to measure tumour growth. Consistent with our *in vitro* results, cells expressing TEAD4-S developed smaller tumours as compared with cells with YAP alone, YAP/TEAD4-FL or even the vector control (Fig. 5a), suggesting that TEAD4-S functions as an inhibitor of tumour development. In addition, the xenograft tumours with TEAD4-S developed much slower than cells with YAP, YAP/TEAD4-FL or even vector control (Fig. 5b), further supporting that TEAD4-S inhibits cancer progression *in vivo*. As a positive control, RBM4 also inhibited tumour development *in vivo*, consistent with its role as antitumour splicing factor to promote TEAD-S.

Next, we surgically collected paired non-small cell lung cancer (NSCLC) samples and adjacent normal tissues from seven patients, and measured the relative levels of two TEAD4 isoforms. Compared with the paired normal tissues, the relative mRNA levels of TEAD4-S were significantly reduced in six out of seven primary NSCLC specimens (Fig. 5c), and the TEAD4-S protein was substantially decreased in five out of seven specimens (Fig. 5d), suggesting a general reduction of TEAD4-S expression despite the obvious heterogeneity in different tumour specimens. Intriguingly, in three of the NSCLC specimens (samples 4, 5 and 7), the truncated TEAD4-S protein was the predominant isoform in normal tissues, but was almost completely eliminated in tumours, implying that the AS switch in TEAD4 could play a major role in the tumorigenesis of these patients.

To further evaluate the clinical relevance of TEAD4 splicing in all cancers, we analysed the databases from TCGA consortium that contains various large-scale RNA-seq results from thousands of patients with various tumours[43] (Supplementary Fig. 4a). Strikingly, TEAD4-S levels were significantly altered in 7 out of 11 tumour types analysed with reduction in the majority of tumours (6 out of 7; Fig. 5e), consistent with our results in the antitumour activity of TEAD4-S. To further investigate the clinical significance of TEAD4 splicing in cancers, we used a survival analysis tool, Kaplan–Meier plotter, to analyse TCGA data sets for the overall survival of various cancer patients with different TEAD4-S levels. Strikingly, a higher level of TEAD4-S was significantly associated with the improved overall survival in patients with lung and colon cancers (Fig. 5f), and to a lesser extent, in patients with liver and breast cancers (Supplementary Fig. 4b, positive but non-significant association). These observations indicate that TEAD4-S could be recognized as an independent prognostic factor for the survival of cancer patients. Collectively, this finding validated the mechanistic link between the TEAD4-S isoform and cancer progression, highlighting the importance of this splicing switch of the Hippo–YAP–TEAD pathway in regulating human cancer progression and patient survival.

## Discussion

Extensive splicing misregulation is one of molecular hallmarks of cancer[44–47]; however, the functional implication is far from clear. Here we report a model in which the AS plays a key regulatory role in mediating the activation of Hippo–YAP signals. In the current model, activation of the Hippo pathway by multiple extracellular cues is converged to its main effector YAP, whose phosphorylation leads to cytoplasmic retention and protein degradation. When unphosphorylated, YAP translocates into the nucleus and interacts with transcription factors TEAD1–4 to activate gene expression and promote cell proliferation[10]. We demonstrated that the AS imposes a new layer of regulation: skipping of exon 3 in TEAD4 produces a short isoform that interacts and neutralizes YAP in both the nucleus and cytoplasm, leading to the attenuation of YAP signalling (Fig. 4g). The splicing of TEAD4-S is controlled by RBM4 through direct binding to its pre-mRNA. Consistently, RBM4 and TEAD4-S inhibit tumour progression in cultured cells and xenograft tumours. Altogether, these data represent a new mechanism on how the AS affects tumorigenesis through mediating the key signalling cascades such as the Hippo–YAP pathway.

Genomic analyses of TCGA data sets indicate that splicing of TEAD4 is commonly altered in cancer patients to reduce TEAD4-S. Strikingly, re-expression of TEAD4-S significantly reduced tumour development in cultured cells and mouse model. Since the Hippo–YAP pathway has a broad impact on

**Figure 2 | Antagonistic function of the two TEAD4 isoforms. (a)** A luciferase reporter driven by CTGF promoter was co-transfected in the presence or absence of YAP as indicated with TEAD4-FL, TEAD4-S or RBM4. The relative luciferase activities were determined by calculating the ratio of firefly luciferase activities over Renilla luciferase activities. Three independent experiments were conducted, with the mean ± s.d. of relative luciferase activities were shown. **(b)** Expression vector of HA-YAP was co-transfected into 293T cells with Flag-TEAD4-FL, Flag-TEAD-S or Flag-TEAD4-FL in the presence of untagged TEAD4-S, and the binding of HA-YAP was determined by co-immunoprecipitation (co-IP) experiment using anti-Flag antibody. **(c)** 293T cells were co-transfected with Flag-TEAD4-FL, HA-YAP and increasing amounts of untagged TEAD4-S. The interaction between HA-YAP and Flag-TEAD4-FL was determined by co-IP assay. The relative levels of two TEAD4 isoforms in these cells were determined by western blot using anti-TEAD4 antibody. **(d)** H157 cells stably expressing TEAD4-FL or TEAD4-S were generated. Chromatin immunoprecipitation (ChIP) from these cells was performed with control IgG and TEAD4 antibody. The precipitation of CTGF promoter was examined by PCR. **(e,f)** H157 and A549 cells stably expressing YAP, YAP/TEAD4-FL, YAP/TEAD4-S, YAP/RBM4 and control were generated. The expression of two YAP target genes, Ctgf and Itgb, was measured by real-time PCR. The mean ± s.d. of relative mRNA levels from triplicate experiments were plotted. **(g)** The transcriptomes of H157 cells expressing YAP, YAP/TEAD4-FL, YAP/TEAD4-S or YAP/RBM4 were determined by RNA-seq. Using hierarchical clustering of all samples, we identified a subcluster of 429 genes that are activated by YAP/TEAD4-FL, but the activation is eliminated by co-expression of YAP with TEAD4-S or RBM4. Relative expression levels are displayed using Java TreeView. **(h)** The protein interaction networks of 429 genes were identified using the STRING database, and the highly connected groups were defined by MCODE and displayed using Cytoscape. The hub proteins interacting with multiple clusters were coded with multiple colours, and the enriched functions/gene ontology terms were labelled.

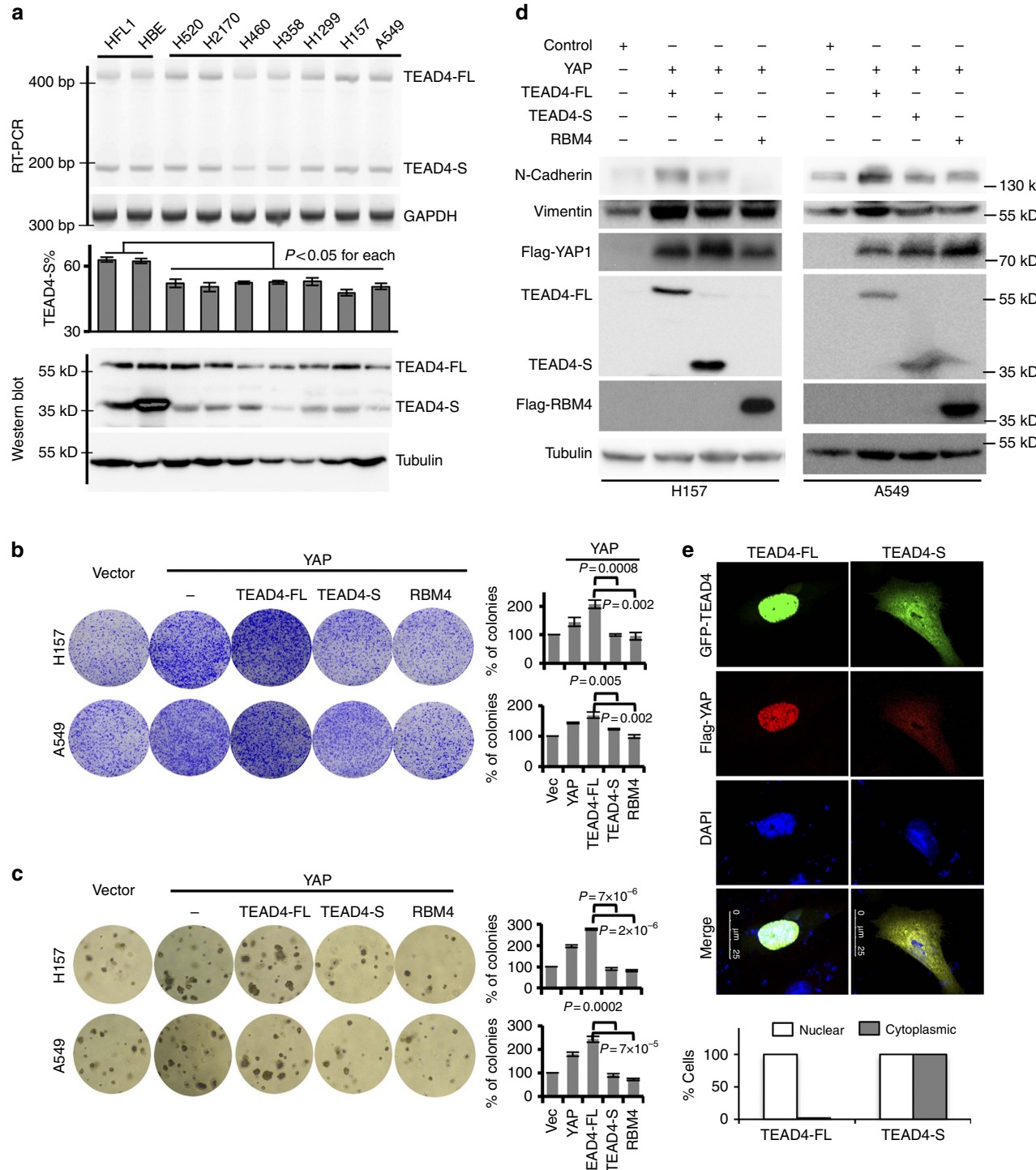

**Figure 3 | TEAD4-S reduces cancer cell proliferation and EMT.** (**a**) RNA levels (upper panel) and protein levels (lower panel) of TEAD4-S in various NSCLC lines and normal bronchial cells were measured by semi-quantitative RT–PCR and western blot. The mean ± s.d. for the percentage of TEAD4-S mRNA was plotted from three experiments. Each cancer cell line was compared with the two normal cell lines to calculate *P* values using Student's *t*-test. (**b**) H157 or A549 cells stably expressing YAP, YAP/TEAD4-FL, YAP/TEAD4-S, YAP/RBM4 or vector control were analysed by colony formation assay. All experiments were performed in triplicates, with mean ± s.d. of relative colony numbers plotted (*P* values from *t*-test). Images of the whole plate were shown. (**c**) H157 or A549 cells stably expressing YAP, YAP/TEAD4-FL, YAP/TEAD4-S, YAP/RBM4 or vector control were analysed by soft agar assay. All experiments were performed in triplicates, with mean ± s.d. of relative colony numbers plotted (*P* values from *t*-test). (**d**) Western blot of epithelial and mesenchymal markers was performed using lysates from indicated H157 and A549 stable cells. (**e**) The subcellular localizations of TEAD4-FL or TEAD4-S with YAP. Cells were co-transfected with Flag-tagged YAP (red) and GFP-fused TEAD4 (green), and visualized with immunofluorescence assay. Scale bar, 25 μm. The localization of TEAD4-FL and TAD4-S was quantified in transfected cells, and the per cent of cells with nuclear or cytoplasmic TEAD4 were plotted (>50 cells were captured and quantified in both samples).

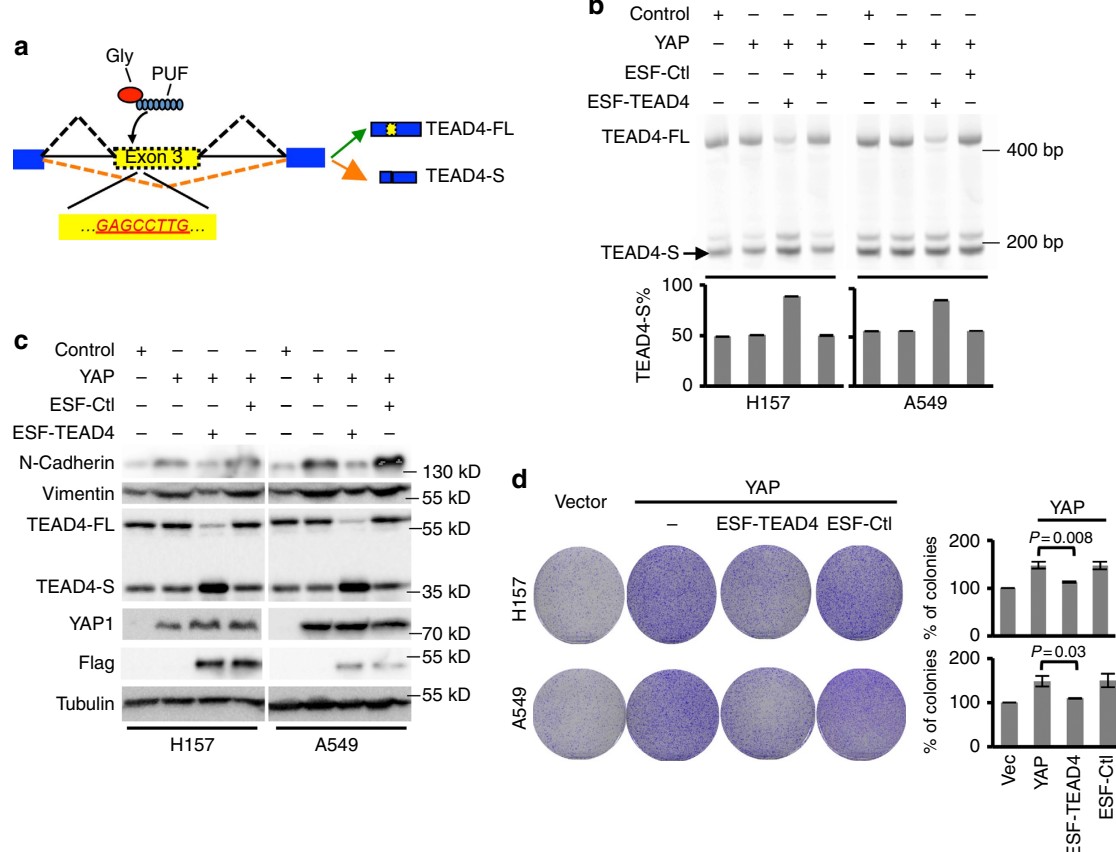

**Figure 4 | The splicing switch to TEAD4-S inhibits EMT and cancer cell proliferation.** (**a**) Schematics of the engineered splicing factor (ESF-TEAD4) that regulates the splicing of TEAD4. PUF domain was engineered to recognize an 8-nt RNA sequence (GAGCCTTG) in exon 3 of TEAD4, and fused with a glycine-rich motif of hnRNP A1 that can inhibit exon inclusion on binding to pre-mRNA. The resulting ESF, ESF-TEAD4, was designed to bind and inhibit the inclusion of exon 3 in TEAD4, thus inducing splicing switch towards TEAD4-S. (**b**) ESF-TEAD4 or control ESF was transfected into YAP-expressing H157 and A549 cells. RNAs were extracted from the transfected cells, and the splicing of TEAD4 was examined with RT–PCR. The representative gel was shown. The mean ± s.d. for the percentage of TEAD4-S mRNA was plotted from three experiments. (**c**) Total proteins were purified from the same set of transfected cells as in **b**, and the epithelial and mesenchymal markers were detected with western blots. (**d**) ESF-TEAD4 or control ESF was transfected into YAP-expressing H157 or A549 cells. Colony formation assay was carried out to examine tumour cell proliferation. All experiments were performed in triplicates, with mean ± s.d. of relative colony numbers plotted (P values from t-test). Images of the whole plate were shown.

tumorigenesis and cancer development, modulating TEAD splicing may provide a new approach for potential therapeutic interventions of cancer. In particular, TEAD4-S is controlled by RBM4, a master regulator of many cancer-related splicing events. Activation of RBM4 increases the production of TEAD4-S, which in turn may potentially inhibit tumorigenesis through multiple oncogenic pathways.

We also noticed that the mRNA and protein levels of two TEAD4 isoforms are not always consistent across different cell lines and tissues (Figs 3a; and 5c,d), suggesting that the two isoforms may be differentially controlled in the levels of protein translation and/or degradation. This observation adds additional layer of complexity in controlling TEAD4 isoforms in addition to splicing regulation at RNA level. Adding to the complexity is the YAP paralogue TAZ that also binds TEAD4 with similar affinity (reviewed in ref. 48). We speculate that the two TEAD4 isoforms may also differentially regulate TAZ-mediated branch of the Hippo pathway; however, an additional research is required to reveal the detailed mechanisms.

Human genome has four TEAD paralogues, TEAD1–4, all of which can interact with YAP to promote transcription. The N-terminal DNA-binding domains of all TEAD proteins are highly conserved. On the basis of the sequence annotation,

TEAD2 and TEAD3 also have putative splicing variants with a truncated N-terminal domain that in principal act similarly to TEAD4-S (Supplementary Fig. 5), although it is unclear whether these splicing isoforms are indeed translated. Our observations that TEAD4-S functions as a dominant negative isoform to sequester/neutralize YAP imply that it may also antagonize all other TEAD paralogues. It is possible that this finding presents a general regulatory switch of all TEAD proteins, and future studies are warranted to fully address the role of all noncanonical TEAD isoforms.

Multiple AS isoforms can also be found in other main components of the Hippo–YAP pathway to differentially affect cellular signals. For example, by skipping the exon 4, YAP can produce a splicing isoform containing single WW domain (YAP1-1). In contrast to the canonical isoform YAP1-2 that contains two WW domains, the YAP1-1 does not bind angiomotin and thus is not sequestered by angiomotin in the cytoplasm[49], In addition, YAP1-1 does not interact with p73 that is the functional partner of YAP1-2 in response to the stress of ultraviolet or serum depravation[50]. Our finding provides a mechanistic link between Hippo signalling and AS regulation, which may be a tip of the iceberg for a general regulatory mode. Previously, the Hippo–YAP pathway is well known to be

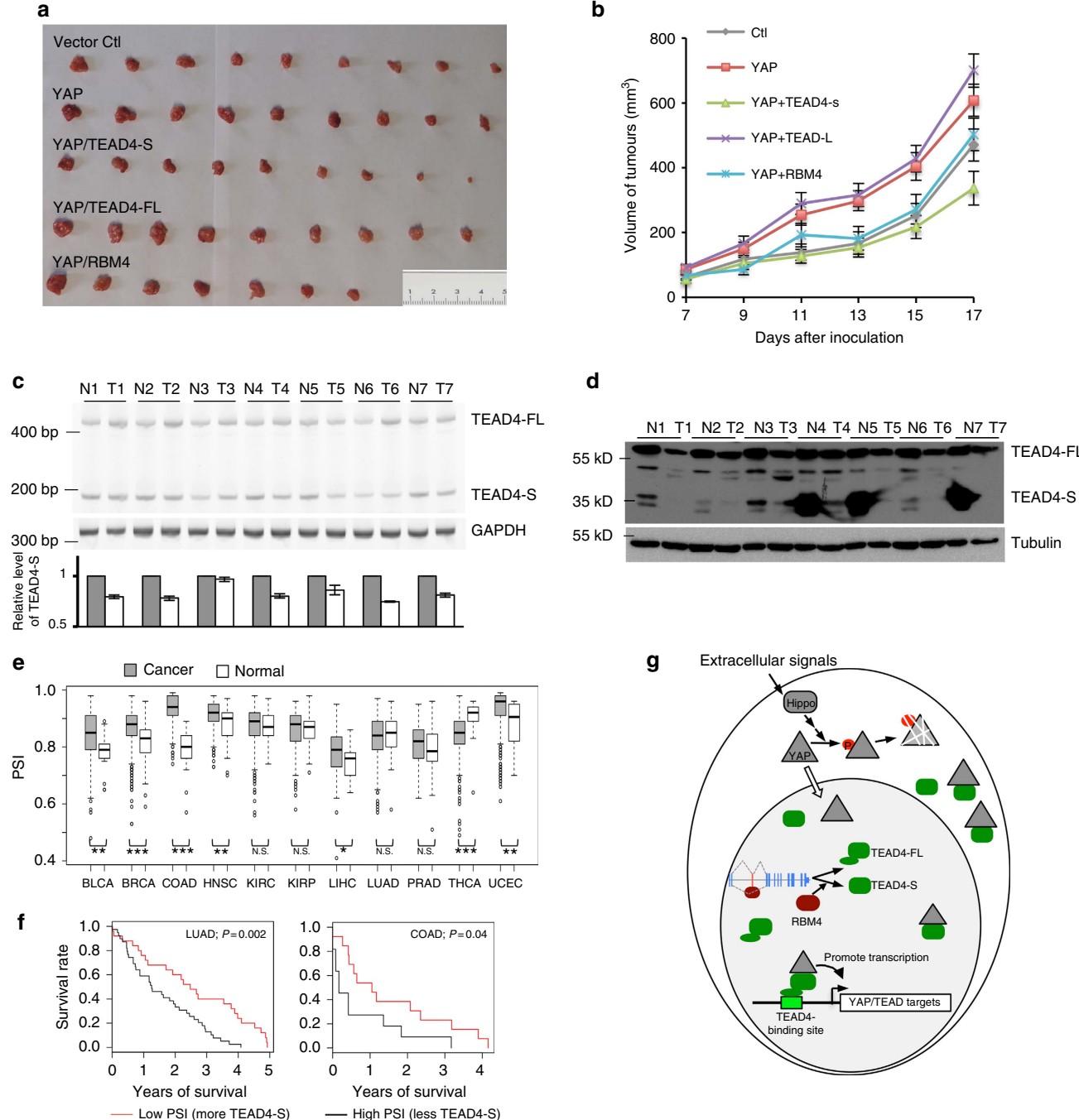

**Figure 5 | TEAD4-S can inhibit tumour growth *in vivo*. (a)** H157 cells stably expressing YAP, YAP/TEAD4-FL, YAP/TEAD4-S, YAP/RBM4 or vector control were subcutaneously injected into the flank of nude mice. Each group contained nine mice. Pictures of the tumours removed after 17 days were shown. Scale bar, 1 cm. **(b)** The growth of xenograft tumours was monitored and the average sizes of xenograft tumours were measured every 2 days ($n = 9$, error bars indicate ± s.d., $P < 0.05$ by $t$-test). **(c)** Total RNA isolated from seven paired NSCLC tumours and adjacent normal tissues were assayed by semi-quantitative RT–PCR. The mean ± s.d. from three experiments was plotted. **(d)** TEAD4 levels from seven paired NSCLC tumours and adjacent normal tissues were analysed by western blot. **(e)** The splicing alteration of TEAD4 was examined in various cancers by analysing the TCGA consortium containing RNA-seq data sets from thousands of patients. The PSI of TEAD4 in the cancer or normal samples are presented as standard box plot, with the boxes presenting the first and third quartiles and the whiskers representing the 2nd and 98th percentiles. \*\*\*$P < 0.0005$; \*\*$P < 0.005$; \*$P < 0.05$; and NS, not significant. **(f)** The overall survivals of lung cancer and colon cancer patients with different ratios of TEAD4 splicing isoforms were analysed. **(g)** The model of how the Hippo–YAP pathway is regulated through a TEAD4 splicing switch.

controlled by a phosphorylation cascade that elicits rapid regulation, the new control mode through splicing may present a slower but more stable regulatory dynamics that is important to cancer cell reprogramming. We speculate that splicing misregulations of other components in the Hippo–YAP pathway also play critical roles in cancer development, and thus should be explored as a new route of potential cancer therapy.

## Methods

**Splicing reporter constructs.** To construct the TEAD4 reporter, we used PCR reactions to amplify a fragment containing exon 2, part of intron 2, exon 3, part of intron 3 and exon 4 of TEAD4, and ligated this fragment to the pCDNA3-FLAG vector digested with NheI/NotI. To generate TEAD4 reporters with mutated RBM4-binding sites, Quikchange approach was applied with different paired primers. All primers used in this study were listed in Supplementary Table 1.

**ESF expression constructs.** To express ESFs in cultured cells, we generated expression constructs using the pCI-neo vector (Promega). We started with an expression construct that encodes from N- to C terminal, FLAG epitope, Gly-rich domain of hnRNP A1 (residues 195–320 of NP_002127), and the MS2 coat protein (gift of Dr R. Breathnach form Institut de Biologie-CHR 1). The fragment encoding the MS2 coat protein fragment was removed using BamHI/SalI digestion and replaced with a fragment encoding a NLS (PPKKKRKV) and the PUF domain of human Pumilio1, which was amplified using primers Pum-F1 and Pum-R1 (Supplementary Table 1). The resulting construct expresses a Gly-PUF-type ESF under the control of a cytomegalovirus (CMV) promoter. To make ESF-TEAD4, we introduced the point mutations in consecutive steps in the PUF domain to make it recognize an 8-nt RNA sequence (GAGCCTTG) using a QuikChange Site-Directed Mutagenesis kit (Stratagene) following the manufacturer's instructions.

**Human tissue total protein and cDNA panels.** Human Tissue total protein and complementary DNA (cDNA) panels were purchased from Amsbio Company. Each panel contained 10 different human tissues, including the brain, colon, liver, kidney, heart, lung, skeletal muscles, pancreas, spleen and stomach.

**Cell culture and transfection.** 293 FlpIn/T-Rex cells were purchased from Invitrogen. All other cell lines were purchased from American Type Culture Collection (Manassas, VA, USA). The HEK293T human embryonic kidney, HeLa cervical cancer, 293 FlpIn/T-Rex and PANC-1 human pancreatic carcinoma cell lines were cultured in DMEM (high glucose) medium containing 10% fetal bovine serum (FBS; Hyclone) and 1% penicillin/streptomycin (P/S). The A549 human lung carcinoma cell line was cultured in F-12 K medium containing 10% FBS and 1% P/S. The H157 human lung carcinoma cell line was cultured in RPMI-1640 medium containing 10% FBS and 1% P/S. The HT-1080 human fibrosarcoma and HepG2 human hepatocellular carcinoma cell lines were cultured in Eagle's Minimum Essential Medium containing 10% FBS and 1% P/S. The MDA-MB-231 human breast cancer cell line was cultured in L-15 medium containing 10% FBS and 1% P/S. The cell lines were tested for mycoplasma contamination using the microbiological culture method, and all the lines were free of mycoplasma.

To generate stable cell line expressing RBM4 on tetracycline induction, we used pCDNA5 FRT/TO vector and 293 FlpIn/T-Rex cells (Invitrogen). The FLAG-tagged full-length RBM4 was cloned into the vector, and transfected with pOG44 in 1:9 ratio. The stably integrated cells were selected with $100\,\mu g\,ml^{-1}$ of hygromycin at 2 days after transfection for ~2 weeks to obtain individual colonies. One day before the induction, the cells were transferred to hygromycin-free medium. The inductions were carried out by adding tetracycline to a final concentration of $2\,\mu g\,ml^{-1}$. The induced cells were collected at several time points after induction to extract RNA and protein for further analysis.

To determine the localization of TEAD4-FL or TEAD4-S, HeLa cells were plated onto a coverslip in six-well plates 1 day before transfection. An amount of $1\,\mu g$ of GFP-tagged TEAD4-FL or TEAD4-S vectors was transfected using lipofectamine 2000 according to the manual. After 48 h, cells were fixed for further immunofluorescence analysis.

To stably express RBM4 (or other proteins) in PANC1 cells (or other cells), we used lentiviral vectors. We transfected 293 cells with pCDH-flag-RBM4 or pCDH-flag-empty vectors as per the manufacture's protocols. The supernatant media containing virus was collected by centrifugation to remove any cellular contaminant. The resulting viral particles were used to infect H157 cells, and stably integrated cells were selected by $5\,\mu g\,ml^{-1}$ of puromycin for 1 week. The expression of transgenes was confirmed by western blots before further analysis.

To determine the effect of overexpression of RBM4 on TEAD4 splicing changes, $0.2\,\mu g$ of TEAD4 mini-gene reporters was co-tranfected with $0.4\,\mu g$ of RBM4, using lipofectamine 2000 according to the manual. After 48 h, cells were collected for further analysis of RNA and protein levels.

**Assay of splicing with semi-quantitative RT–PCR.** The total RNA was isolated from transfected cells with TRIzol reagent (Invitrogen) according to the manufacturer's instructions, followed by 1-h DNase I (Invitrogen) treatment at $37\,°C$, and then heat inactivation of DNase I. Total RNA ($2\,\mu g$) was then reverse-transcribed with SuperScript III (Invitrogen) using poly T primer, and one-tenth of the room temperature product was used as the template for PCR amplification (25 cycles of amplification, with trace amount of Cy5-dCTP in addition to non-fluorescent dNTPs). Reverse transcription–PCR (RT–PCR) products were separated on 10% polyacrylamide gel electrophoresis (PAGE) gels, and scanned using a Typhoon 9400 scanner (Amersham Biosciences). The amount of each splicing isoform was measured using ImageQuant 5.2.

**Western blot.** Cells were lysed in lysis buffer containing 50 mM HEPES, 150 mN NaCl (4.38 g), 1 mM EDTA, 1% (w/v) CHAPS and Sigma protease inhibitor cocktail. Subsequently, the cell lysates were boiled in $2\times$ SDS–PAGE loading buffer for 10 min, and then resolved by 10% SDS–PAGE and transferred to the nitrocellulose membrane. All primary antibodies were diluted 1,000 times for western blotting if not specified. The following antibodies were used in this study: TEAD4 (#ab58310) antibody, anti-Myc tag antibody (#ab9106) and anti-HA tag antibody (#ab9110) were purchased from Abcam; N-cadherin (#610921) antibody was purchase from BD; Vimentin (#5741) and YAP (#12395) antibodies were purchased from Cell Signaling Technology; and alpha-tubulin (#T5168, 1:5000 dilution) and FLAG M2 (#F1804) were purchased from Sigma-Aldrich. RBM4 antibody (#11614-1-AP) was purchased from Proteintech. Bound antibodies were visualized using the ECL kit (GE Healthcare).

**Clinical tissues samples collection.** Fresh lung cancer tissues and normal adjacent tissues were collected from patients with pathologically and clinically confirmed lung carcinomas. All human tumour tissues were obtained with written informed consent from patients or their guardians before participation in the study. The Institutional Review Board of the Dalian Medical University approved use of the tumour specimens in this study. All of tissue specimens were kept in liquid nitrogen and sectioned for protein or mRNA extraction.

**High-throughput mRNA-sequence and data analysis.** RNAs from H157 cells stably expressing YAP, YAP/TEAD4-FL, YAP/TEAD4-S, YAP/RBM4 and control vectors were purified using Trizol method, and subsequently cleaned using the RNAeasy Kit (Qiagen). The RNAs were digested in column with RNAse-free DNAse as per the manufacturer's instruction. Total RNA not exceeding $3\,\mu g$ was further used to purify polyadenylated RNA using the Illumina TruSeq Total RNA Sample Prep kits. We used the Ribo-Zero Human to remove the cytoplasmic rRNA. The mRNA purified was further analysed using the Bioanalyzer (Agilent Technologies) before generation of cDNA library with bar-coded ends. RNA-seq libraries were robotically prepared using the Illumina TruSeq Total RNA Sample Prep kits according to the manufacturer's protocol. The RNA-seq data set was deposited to the Gene Expression Omnibus with accession number GSE80372.

To estimate the gene expression levels, we used RSEM package and bowtie2 (refs 51,52) to align all reads to human reference genome (UCSC hg19 version). Then, we provided a fragment length distribution with options of '–fragment-length-mean 75' and '–fragment-length-sd 10' to calculate transcript expression levels. Subsequently, we used EBSeq tool[53] to examine differential expression genes of pair-wise comparison based on empirical Bayesian methods.

**Heat map.** We included genes that met the following criteria: (i) FPKM (fragments per kilobase of transcript per million mapped reads) values of a given gene are not equal in all samples; (ii) at least one of the FPKM values in all samples is $\geq 3$; and (iii) the ratio of the maximum FPKM value and the minimum FPKM value in all samples is $\geq 2$. Then, we used the log 2 ratio of FPKM values of the included genes, normalized by the FPKM value of control sample, as input of Cluster 3.0 (ref. 54). We clustered the data set using the hierarchical clustering method based on Pearson correlation with average linkage, and further viewed the results using Java TreeView.

We selected the cluster in this pattern—upregulated by YAP–TEAD434, but downregulated by YAP–TEAD305 and YAP–RBM4—as our target data set. This data set includes the genes that are differentially regulated by TEAD4 isoforms. The heat map shown is ordered by the FPKM value of YAP.

The gene ontology analysis was performed using DAVID gene ontology analysis software to search for enriched pathways. The functional association of TEAD4 targets were analysed using the protein interaction data from STRING database, generating a set of functional interaction networks. The sub-network containing more than five nodes were demonstrated.

**Soft agar assay.** Equal volumes of 1.2% agar and $2\times$ DMEM (or RPMI-1640) mediums were mixed and placed onto six-well dishes to generate 0.6% base agar. A549 cells (or H157 cells) expressing YAP, YAP/TEAD4-FL, YAP/TEAD4-S, YAP/RBM4 and control vectors were seeded in 0.3% top agar ($10^4$ cells per plate) and incubated at $37\,°C$ in humidified atmosphere for 3 weeks. Colonies were stained with crystal violet and counted.

**Colony formation assay.** A549 cells (or H157 cells) expressing YAP, YAP/TEAD4-FL, YAP/TEAD4-S, YAP/RBM4 and control vectors (5,000 cells per dish) were seeded in the 10-cm dishes and incubated at $37\,°C$ in humidified incubator for 2 weeks. Colonies were fixed and stained with crystal violet, and the number of colonies was counted.

**Xenograft assays.** Forty-five 4-week-old female BALB/c nude mice were purchased from Vital River Laboratories for in vivo tumorigenicity study. The Institutional Animal Care and Use Committee of the Dalian Medical University approved the use of animal models in this study. Mice were injected

subcutaneously with $1 \times 10^6$ H157 cells expressing YAP, YAP/TEAD4-FL, YAP/TEAD4-S, YAP/RBM4 and control. Nine mice were used for each group. Mice were raised in the following 3 weeks. The mice were then monitored for tumour volume and overall health. The size of the tumour was determined by caliper measurement of the subcutaneous tumour mass every 3 days. Tumour volume was calculated according to the formula $(4/3)\pi r_1^2 r_2$, $(r_1 < r_2)$. Each experimental group contained nine mice. At the end of 17 days, all mice were killed, and tumours were removed for further analysis. For all data points, three independent measurements were performed and means were used for calculation.

**RNA immunoprecipitation.** 293T cells ($1 \times 10^6$) expressing RBM4 or control vector are collected and washed twice with 10 ml of PBS, and then resuspended in 10 ml of PBS. Formaldehyde (37% stock) is added to the above solution to a final concentration of 1% and incubated at room temperature for 10 min with slow rotating. Crosslinking reactions are quenched by the addition of glycine solution (pH 7.0) to a final concentration of 0.25 M, followed by incubation at room temperature for 5 min. The cells are collected by centrifugation at 700g for 4 min at 4 °C, followed by two washes with ice-cold PBS. Fixed cells are resuspended in 2 ml of radioimmunoprecipitation assay (RIPA) buffer (50 mM Tris-Cl, pH 7.5, 1% NP-40, 0.5% sodium deoxycholate, 0.05% SDS, 1 mM EDTA, 150 mM NaCl) containing protease inhibitors. The cells are subsequently lysed by three rounds of sonication. Insoluble material is removed by microcentrifugation at 16,000g for 10 min at 4 °C. An aliquot of solubilized cell lysate is mixed with protein A–Sepharose beads along with nonspecific competitor tRNA. This mixture is rotated for 1 h at 4 °C, followed by microcentrifugation at 1200g for 5 min. The supernatant is removed and used for immunoprecipitation.

Protein A or protein G–Sepharose beads are coated with the Flag antibody for 2 h at 4 °C, followed by extensive washing with RIPA buffer containing protease inhibitors. Before immunoprecipitation, the beads are incubated for 10 min in RNasin. The precleared lysate is diluted with RIPA buffer, mixed with the antibody-coated beads and incubated with rotation for 60–90 min. The beads are collected using a minicentrifuge at 2800g for 45 s. The beads are washed five or six times with 1 ml of highstringency RIPA buffer (50 mM Tris-Cl, pH 7.5, 1% NP-40, 1% sodium deoxycholate, 0.1% SDS, 1 mM EDTA, 1 M NaCl, 1 M urea, 0.2 mM phenylmethyl sulphonyl fluoride) by 10-min rotation at room temperature. The beads containing the immunoprecipitated samples are collected and resuspended in 100 μl of 50 mM Tris-Cl, pH 7.0, 5 mM EDTA, 10 mM dithiothreitol and 1% SDS. Samples (resuspended beads) are incubated at 70 °C for 45 min to reverse the crosslinks. The RNA is extracted from these samples using Trizol according to the manufacturer's protocol, and reverse-transcribed into cDNA for PCR detection.

**Immunofluorescence staining.** To determine the localization of TEAD4-FL, TEAD4-S and YAP, we performed immunofluorescence assay. In brief, cells were plated on coverslips to appropriate density. Transfected cells were fixed on the coverslips with 4% paraformaldehyde in 1× PBS for 15 min at room temperature and washed with 1× PBS three times. Cells were then permeabilized with 0.2% Triton X-100 for 10 min. After blocking in 3% bovine serum albumin for 30 min, slides were incubated with indicated antibodies (Flag or Myc, 1:100 dilution) antibody diluted in 1% bovine serum albumin for 2 h. Subsequently, slides were washed with 1× PBS for three times, and then incubated with fluorophore-conjugated secondary antibodies for 1 h. The coverslips were then washed and mounted with mounting medium (Vector shield's mounting medium with 4,6-diamidino-2-phenylindole). Cells were visualized using an Olympus fluorescence microscope, and photographs were generated using a Kodak digital camera.

**Luciferase reporter assay.** For the luciferase reporter assay, HEK293T cells were seeded in 24-well plates. Cells were co-transfected with 100 ng of CTGF promoter/firefly luciferase reporter plasmid and different amount of YAP (100 ng), TEAD4-FL (100 ng), TEAD4-S (20 or 100 ng), 100 ng TEAD4-FL with increasing amounts of TEAD4-S (20, 50 and 100 ng) and RBM4 (100 ng) plasmids and 5 ng of pRL-TK Renilla plasmids using lipofectamine 2000 (Invitrogen). After 48 h of transfection, cells were either lysed in protein lysis buffer for protein extraction or in passive lysis buffer (Promega) for luciferase assay measured with the Dual-Luciferase Reporter Assay System (Promega), using the TD-20/20 Luminometer (Turner Designs). The relative luciferase activities were determined by calculating the ratio of firefly luciferase activities over Renilla luciferase activities.

**Assay of CTGF and ITGB expression with real-time PCR.** The real-time PCR was performed using the Maxima SYBR Green qPCR Master Mix (Thermo Scientific) and a 7500 real-time PCR system (Life Technologies) according to the manufacturer's instructions. The expression level of CTGF and ITGB was normalized to the endogenous expression of GAPDH.

**Statistics.** Statistical analyses of colony formation, soft agar and splicing changes were performed using Student's *t*-test.

**Data availability.** RNA-seq data that support the findings of this study have been deposited in Gene Expression Omnibus of NCBI with the accession code GSE80372. The authors declare that all the data supporting the findings of this study are available within the article and its Supplementary Information files.

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

## Acknowledgements

We thank Drs Woan-Yuh Tarn and Kun-liang Guan for providing reagents. We thank Dr Xiaoling Li for critical reading of manuscript. This work is supported by the National Natural Science Foundation of China (31471235, 81422038 and 91540110 to Y.W.; 31570823 to Z.W.; 31400726 to W.Z.), the NIH grant R01CA158283 (to Z.W.), Young Thousand Talents Program of China (to Y.W.), the Education Department of Liaoning Province in China (the 'Program for Pan-Deng Scholars' to Y.W.), and program for Chang Jiang Scholars and Innovative Research Team in University IRT13049.

## Author contributions

Y.W. and Z.W. designed the experiments, interpreted the results and wrote the manuscript. Y.Q., W.H., H.Q., J.Z., H.W. and W.Z. performed the experiments. Q.L. and S.M. help to interpret the data. J.Y., X.F. and Y.T. analysed the RNA-seq and TCGA data.

## Additional information

**Accession codes:** RNA-seq data that support the findings of this study have been deposited in Gene Expression Omnibus of NCBI with the accession code GSE80372.

**Competing financial interests:** The authors declare no competing financial interests.

