## [Peer Review File · Nature Communications]

Reviewer #1 (Remarks to the Author):Expert in RNA alternative splicing

The authors identify an alternatively spliced isoform of TEAD4 and propose that it acts as a dominant negative against the full-length form as transcriptional coactivator with YAP. The results support the expression of the short isoform at least in cell lines and the that RBM4 regulates splicing of TEAD4. The authors used minigenes to define the RBM4 binding site mediating its function. As noted below, there are some issues with the conclusions. Specifically, while the results from studies overexpression different combinations of proteins are consistent with a role for the short TEAD4 isoform in suppressing YAP signaling and tumor growth, the results do not definitively link the TEAD4 short form with the effects.

The studies in Fig 2 do not convincingly show that the short isoform functions as a dominant negative. While the short form does not have transcriptional activity in Fig 2a, the authors need to show that co expression of the short form inhibits transcriptional activity of the full length form without reduce the expression level of the full length form.

The claim that expression of RBM4 controls YAP signaling via splicing of TEAD4 is not convincingly demonstrated in Fig. 2. Given that RBM4 overexpression could have multiple effects that produces decreased expression of the two TEAD4 target genes chosen, a stronger connection with TEAD needs to be established. Is it possible to use an antisense oligo to block binding of RBM4 and block it's effect on TEAD splicing so that the full length isoform is expressed in the presence of RBM4, for example? What is the effect of RBM4 expression without YAP; if it reduces transcription below the baseline, its effect is likely to be TEAD4-independent. Similarly, the experiment shown in Fig 2f could use an RBM4 alone control.

Also regarding Fig. 2f, It is not clear in this RNA-seq experiment what level of overlap there is between YAP/TEAD-S and YAP/RBM4. If RBM4 is mediating the altered expression by generating the endogenous TEAD-S, there should be substantial overlap in the target genes affected.

The experiments to demonstrate an effect of the TEAD4-S isoform in cancer vs normal cells would be much more convincing without overexpression studies. To show that the proliferation and EMT phenotypes of the cancer cell lines shown in Fig 3a are dependent specifically on TEAD4-S, the authors can use oligos to block inclusion of exon 3 to force a switch from the full length to the short isoform. This will provide a direct demonstrate of the requirement for only this isoform.

Reviewer #2 (Remarks to the Author): Expert in Hippo pathway

A-H in various order

Yangfan Qi and colleagues from Zefeng Wang laboratory uncover a novel mechanism of YAP regulation that occurs via alternately spliced isoform of TEAD4 (TEAD4-S), which acts as a dominant negative regulator of YAP-TEAD signaling. The authors show that the splicing factor RBM4 increases the abundance of TEAD4-S isoform via interaction with TEAD4 mRNA and demonstrate that overexpression of TEAD4-S or RBM4 have similar effects on YAP driven gene expression, proliferation, and EMT.

The study is well designed it is thorough and novel. The authors do not try to expand their study to investigate splicing of the remaining TEAD isoforms (1-3), though this is touched on in their Discussion, rather they soundly characterize the role of RBM4 splicing factor in mediating YAP-TEAD4 driven gene transcription and biology (proliferation and EMT) in multiple cell lines and using several different methods. They have performed numerous biochemical analyses as well as include good supporting clinical data from human patient samples as well as in vivo mouse studies of tumor biology.

The following changes are suggested to improve the manuscript:

Major Points:

On page 3, the sentence in Introduction section claiming that "all components of Hippo-YAP pathway undergo extensive regulation in RNA level through alternative splicing (AS)" is inaccurate and should either be changed to state that SOME components of Hippo-YAP pathway undergo alternative splicing, or should be referenced accordingly.

It would be helpful for the authors to include more detail regarding the putative RBM4 binding recognition sequence present in TEAD4 and the mutant sequences generated (mut1 and mut2), as it is not clear from Figure 1f. The authors could also discuss whether this sequence is present in other components of the Hippo signaling pathway that undergo alternate splicing e.g., YAP itself.

YAP isoforms have been shown to signal differentially, for example, YAP 1-1 with one WW domain does not bind to Angiomotin (AMOT) and its is not sequestered by AMOT in the cytoplasm in contrast to YAP1-2, with two WW domains. Also YAP 1-1 does not interact with p73, the functional partner of YAP1-2 in response to stress of UV or serum deprivation. These examples of functional differences among YAP isoforms could be discussed [refs: Oka, T., et al., (2008) J. Biol. Chem. 283, 27534-27546. And Oka, T., et al., (2012)]

Significant disparity between mRNA and protein expression levels for TEAD4-FL/-S (Figures 3a, and 4c/d) should be addressed in the Discussion section.

Minor Points

Supplementary Figure 2a is not referenced in the Results section.

The authors should specify more clearly in the Results section and relevant Figure legends when blotting for TEAD4 whether FLAG-tagged or endogenous protein has been assessed, e.g., Figure 1b.

A brief mention of TAZ, a paralog of YAP, and its potentially different mode of interaction with TEAD is suggested in the discussion.

Overall clarity of the manuscript could be improved by further editing for grammar and sentence structure.

Reviewer #3 (Remarks to the Author): Expert in Hippo signalling

The manuscript by Qi et al describes a new TEAD4 splicing variant that lacks exon 3. This variant lacks the DNA binding domain and thus cannot promote gene expression but retains the ability to interact with YAP. The authors thus propose that this short variant acts as a dominant negative to compete with full-length TEAD4 and inhibit YAP induced gene expression. RBM4 is identified as the splicing factor that promotes exon skipping in TEAD4-S. The authors go on to show that as compared to TEAD4-FL, TEAD4-S is not able to promote cell proliferation, colony formation on soft agar, expression of EMT markers or promote in vivo tumor formation in the context of YAP overexpression. Analysis of cancer databases shows a correlation between TEAD4-S expression levels and overall cancer patient survival at least in lung and colon cancers. The main conclusion drawn by the authors is that the ability of TEAD-S to block YAP function in normal cells is lost due to a decrease in expression of TEAD4-S and this promotes the YAP-induced tumorigenic phenotype. However, while the authors have shown that TEAD-S is incapable of cooperating with overexpressed YAP, insufficient evidence is provided to support the contention that TEAD4-S normally functions to repress YAP activity by competing with endogenous FL YAP and that loss of TEAD4-S in cancers results in enhanced YAP signalling. Specific comments are provided below.

1. The majority of the experiments compare the effect of overexpressing TEAD FL versus TEAD S on target gene expression, proliferation etc. However, at best the data shows that TEAD S is not able to promote the observed effect. It does not provide evidence that it can compete with FL TEAD to regulate outcome and thus that a loss would increase YAP signalling. The authors need to focus on the effect of TEAD S as compared to Vector alone

2. Given the importance to the study, the authors must describe and show data confirming that the TEAD primers they are using to detect full length and short TEAD are specifically detecting the expected products. There are many ways this could be done (siRNAs, re-PCR of purified DNA with nested primers, cloning and sequencing the PCR product, etc). Neither positive or negative controls are shown in any RT-PCR experiments.

It would also be useful if the authors could explicitly state what region the TEAD antibody they are using was raised against and/or recognizes in TEAD.

3. Figure 1c. What is the tag on the transfected YAP? What antibody was used to detect YAP?

4. In most experiments where RNA and protein levels of TEAD4-FL versus -S are compared, (ie Fig. 1d,)), the relative ratio of TEAD FL to S mRNA does not correlate with that observed in the protein blotting. Is there an explanation?

In this regard, in Figure 3a, the differences in TEAD4-FL and TEAD4-S expression as assessed by RT-PCR between normal cells and cancer cells is not very convincing, although there is a clear difference in protein levels. The same is true in Figure 4c and d.

5. Figure 1f-h. What exactly is mut2? Does this sequence comply with a known RBM consensus site?

6. Figure 2a. For the reporter assay data the authors need to show that the levels of the FL vs S TEAD protein is matched, preferably in an aliquot of the sample subjected to reporter assay analysis.

7. Figure 2b. The experimental design is unclear. Is it an untagged TEAD-S used in lane 4? In this case, the authors should blot using the TEAD antibody to visualize total protein levels, not just

with the Flag antibody that only detects FL. In this competition experiment it is important that the ratio of FL to S is assessed. Did the authors try a titration to support the notion that it is a competition? Did the authors try IP'ing YAP and blotting for the FL and S TEAD as an alternative to show how effective the S form is in competing for binding?

8. Figure 2c-e. The total level of expression of Tead FL vs S and YAP in the stable cell lines must be shown using lysates from cells grown in parallel. In the case of RBM cells, the increase in TEAD S protein should also be shown. A positive control in the case of TEAD S and YAP/RBM samples would also be useful. Do the authors have an explanation for the modest effect of YAP/RBM on Itgb in A549 cells?

9. In regards to the RNA-seq data shown in Figure 2f; YAP is thought to function primarily through cooperation with TEAD, so it is unclear why overexpression of YAP + TEAD should give such a dramatically different set of induced genes. If TEAD-S competes for binding, I would expect that this construct would be more effective against the endogenous TEAD. Have the authors done a detailed analysis of genes downregulated by TEAD-S as compared to empty vector cells or cells expressing YAP alone? Since the authors are arguing the TEAD S can compete with TEAD FL, this would be the most relevant comparison.

Finally, insufficient information is provided to assess the results of the RNAseq. How many genes were increased/decreased in each cell line type (ieYAP, YAP+TEAD-FL, etc) as compared to the vector and/or YAP alone? Are there many genes that are increased in both TEAD FL versus S?

10. Figure 3e. Quantitation of localization is essential. The authors should also confirm antibody specificity by showing images of YAP staining in non-transfected cells.

11. Figure 4d. Three of the tumor samples show a significant difference in the ratio of S to FL, but in all of the others, the tumor sample is underloaded and thus does not support the authors conclusion that there is a change in the levels. The text should be modified accordingly.

12. Supplementary Figure 1a, b. The legends are so brief that it is not possible to easily understand what is being shown. For example, in a, the legend simply says 'mRNA levels'. I suspect this must be a DNA stained gel from an RT-PCR reaction.

13. Suppl. Fig. 2c is not explained in the results, nor does the legend provide enough detail to understand what point is being made.

14. Suppl. Fig. 3. Blots from two cells lines are shown, but the lanes are not labeled to indicate which cell line is which. The tubulin levels for the vector lane for cell line shown on the right are very low, thus the blotting for protein levels, particularly YAP are not convincing.

Reviewers' comments:

Reviewer #1 (Remarks to the Author):Expert in RNA alternative splicing

The authors identify an alternatively spliced isoform of TEAD4 and propose that it acts as a dominant negative against the full-length form as transcriptional coactivator with YAP. The results support the expression of the short isoform at least in cell lines and the that RBM4 regulates splicing of TEAD4. The authors used minigenes to define the RBM4 binding site mediating its function. As noted below, there are some issues with the conclusions. Specifically, while the results from studies overexpression different combinations of proteins are consistent with a role for the short TEAD4 isoform in suppressing YAP signaling and tumor growth, the results do not definitively link the TEAD4 short form with the effects.

We thank the reviewer for the overall positive comments on this work, and have provided new data to strengthen the link of TEAD4-S with the observed effects on suppressing YAP signaling and tumor growth. Briefly, we demonstrated that the short isoform of TEAD4 can inhibit the transcriptional activity of the full length TEAD4 in a dose dependent manner (new Figure 2a, see below in response to the first point of this reviewer). Additionally, we shifted the splicing of TEAD4 towards TEAD4-S using engineered splicing factors (ESFs), which partially reversed the EMT phenotype resulting from YAP. Importantly, such TEAD4 splicing switch indeed inhibited tumor cell proliferation as judged by colony formation assay (new Figure 4, see below for response to the fourth point of this reviewer). Therefore, these data provide direct evidence to support the link of TEAD4 short form with the resulting effects.

1.The studies in Fig 2 do not convincingly show that the short isoform functions as a dominant negative. While the short form does not have transcriptional activity in Fig 2a, the authors need to show that co expression of the short form inhibits transcriptional activity of the full length form without reduce the expression level of the full length form.

The reviewer raised a good point, and we have carried out the experiments based on the suggestion. We co-expressed TEAD4-FL isoforms with increasing amounts of TEAD4-S, and found a dose dependent reduction of the transcriptional activity of TEAD4-FL as judged by the luciferase assay. We have now modified the Fig. 2a to include these data (new Figure 2a and page 7 in the main text, also see below).

Figure 2a. A luciferase reporter driven by CTGF promoter was co-transfected in the presence or absence of YAP as indicated with TEAD4-FL, TEAD4-S, or RBM4. The relative luciferase activities were determined by calculating the ratio of firefly luciferase activities over Renilla luciferase activities. The data showed TEAD4-S can directly inhibit the transcriptional activity of TEAD4-FL in a dose dependent manner.

2. The claim that expression of RBM4 controls YAP signaling via splicing of TEAD4 is not convincingly demonstrated in Fig. 2. Given that RBM4 overexpression could have multiple effects that produces decreased expression of the two TEAD4 target genes chosen, a stronger connection with TEAD4 needs to be established. Is it possible to use and antisense

oligo to block binding of RBM4 and block its effect on TEAD splicing so that the full length isoform is expressed in the presence of RBM4, for example? What is the effect of RBM4 expression without YAP; if it reduces transcription below the baseline, its effect is likely to be TEAD4-independent. Similarly, the experiment shown in Fig 2f could use an RBM4 alone control.

As suggested, we co-transfected antisense oligos with RBM4 expression vector, and found that RBM4 can't modulate the splicing of TEAD4 once the binding site in exon 3 of TEAD4 was blocked (Supplementary Fig. 1g and page 7 in the main text).

Supplementary Fig. 1g. The antisense oligo that masks RBM4 binding site was co-transfected with the RBM4 expression vector or control vector into A549 cells, and the splicing of TEAD4 was determined by RT-PCR. As expected, the data showed that RBM4 can no longer regulate the splicing of TEAD4 once its binding site in exon 3 was blocked.

As suggested, we also examined the effect of RBM4 expression without YAP according to the reviewers' comments, and found that RBM4 itself didn't affect the transcription of CTGF as judged by luciferase assay, suggesting that the effect of RBM4 is indeed dependent on TEAD4 splicing (new Figure 2a, page 8 in main text, compare the first to the last sample).

Figure 2a. A luciferase reporter driven by CTGF promoter was co-transfected in the presence or absence of YAP as indicated with TEAD4-FL, TEAD4-S, or RBM4. The relative luciferase activities were determined by calculating the ratio of firefly luciferase activities over Renilla luciferase activities. The data showed that RBM4 alone didn't affect the transcription of CTGF.

Similarly, the experiment shown in Fig 2f could use an RBM4 alone control.

As suggested, we tested if RBM4 alone could affect the expression of CTGF and ITGB, and found RBM4 didn't affect their expression levels (new Supplementary Fig. 2e and page 8 in the main text, see below).

Supplementary Fig. 2e. The expression of CTGF and ITGB were not affected by RBM4 alone. The expression levels of both genes were determined by realtime RT-PCR in H157 and A549 cells stably transfected with RBM4 or control vector.

3. Also regarding Fig. 2f, It is not clear in this RNA-seq experiment what level of overlap there is between YAP/TEAD-S and YAP/RBM4. If RBM4 is mediating the altered expression by generating the endogenous TEAD-S, there should be substantial overlap in the target genes affected.

We performed the suggested analyses, and found that the genes up-regulated or down-regulated upon YAP/TEAD4-S expression are indeed significantly overlapped with those altered in YAP/RBM4 expression, whereas the genes altered in different directions are mutually exclusive. These data are included as new supplementary Fig. 2h (page 8 in the main text) and below.

Supplementary Fig. 2h. The genes up-regulated or down-regulated upon YAP/TEAD4-S expression are significantly overlapped with those altered in YAP/RBM4 expression. $P < 2.2 \times 10^{-16}$ as judged by chi-square test.

4. The experiments to demonstrate an effect of the TEAD4-S isoform in cancer vs normal cells would be much more convincing without overexpression studies. To show that the proliferation and EMT phenotypes of the cancer cell lines shown in Fig 3a are dependent specifically on TEAD4-S, the authors can use oligos to block inclusion of exon 3 to force a switch from the full length to the short isoform. This will provide a direct demonstrate of the requirement for only this isoform.

This is again a very good point. As suggested, we used a unique approach developed in our lab, engineered splicing factors (ESFs) (Wang Y et. al. Nature Methods 2009), to specifically shift TEAD4 splicing towards the short isoform. Such splicing switch to TEAD4-S indeed partially reversed the EMT phenotypes, and inhibited tumor cell proliferation as judged by colony formation assay. These data were included as a new Figure 4 (page 10 and 11 in the main text, also see below), which provide direct evidence to support the link of TEAD4 short form with the resulting effects. We thank the reviewer for this insightful suggestion.

Figure 4. The splicing switch to TEAD4-S inhibits EMT and cancer cell proliferation. (a) Schematics of the ESF that regulates the splicing of TEAD4. (b) ESF-TEAD4 specifically shifted the splicing of TEAD4 towards the short form. The splicing switch to TEAD4-S inhibits YAP-mediated EMT (c) and cancer cell proliferation (d) in two different lung cancer cell lines. See page 10 and 11 of main text for details.

Reviewer #2 (Remarks to the Author): Expert in Hippo pathway

A-H in various order

Yangfan Qi and colleagues from Zefeng Wang laboratory uncover a novel mechanism of YAP regulation that occurs via alternately spliced isoform of TEAD4 (TEAD4-S), which acts as a dominant negative regulator of YAP-TEAD signaling. The authors show that the splicing factor RBM4 increases the abundance of TEAD4-S isoform via interaction with TEAD4 mRNA and demonstrate that overexpression of TEAD4-S or RBM4 have similar effects on YAP driven gene expression, proliferation, and EMT.

The study is well designed it is thorough and novel. The authors do not try to expand their study to investigate splicing of the remaining TEAD isoforms (1-3), though this is touched on in their Discussion, rather they soundly characterize the role of RBM4 splicing factor in mediating YAP-TEAD4 driven gene transcription and biology (proliferation and EMT) in multiple cell lines and using several different methods. They have performed numerous biochemical analyses as well as include good supporting clinical data from human patient samples as well as in vivo mouse studies of tumor biology.

We thank the positive feedbacks from this reviewer, and appreciate the comments that "the study is well designed it is thorough and novel."

Major Points:

1. On page 3, the sentence in Introduction section claiming that "all components of Hippo-YAP pathway undergo extensive regulation in RNA level through alternative splicing (AS)" is inaccurate and should either be changed to state that SOME components of Hippo-YAP pathway undergo alternative splicing, or should be referenced accordingly.

As suggested, we have changed this sentence to "some" components, and used YAP and MST1 as examples of Hippo-YAP components with multiple splicing isoforms (page 3).

2. It would be helpful for the authors to include more detail regarding the putative RBM4 binding recognition sequence present in TEAD4 and the mutant sequences generated (mut1 and mut2), as it is not clear from Figure 1f. The authors could also discuss whether this sequence is present in other components of the Hippo signaling pathway that undergo alternate splicing e.g., YAP itself.

The putative RBM4 binding sequence (CGGCCGG) and mutation sequences (mut1: CTTATA; mut2: GTAACG) were selected according to our previous work (new citation added in page 6), and we have now included the sequence information in the main text (page 6) and figure legend of Fig 1f for clarity. We did not find the RBM4 binding site in the alternative exons of YAP. Consistently, when testing if RBM4 can affect alternative splicing of YAP1, we found that RBM4 has no effect on the splicing of YAP exon 6 (see below).

Rebuttal figure 1. The splicing of exon 6 of YAP1 was examined in RBM4-expressing or control cells with RT-PCR. RBM4 doesn't affect its splicing.

3. YAP isoforms have been shown to signal differentially, for example, YAP 1-1 with one WW domain does not bind to Angiomotin (AMOT) and its is not sequestered by AMOT in the cytoplasm in contrast to YAP1-2, with two WW domains. Also YAP 1-1 does not interact with p73, the functional partner of YAP1-2 in response to stress of UV or serum deprivation. These examples of functional differences among YAP isoforms could be discussed [refs: Oka, T., et al., (2008) J. Biol. Chem. 283, 27534-27546. And Oka, T., et al., (2012)]

We thank the reviewer for this insight, and have discussed these examples in page

16 of the main text, with relevant references cited as reference 25 and 26.

4. Significant disparity between mRNA and protein expression levels for TEAD4-FL/-S (Figures 3a, and 4c/d) should be addressed in the Discussion section.

We discussed this in page 15: “We also noticed that the mRNA and protein levels of two TEAD4 isoforms are not always consistent across different cell lines and tissues (Fig. 3a and Fig. 5c and 5d), suggesting that the two isoforms may be differentially controlled in the levels of protein translation and/or degradation. This observation adds additional layer of complexity in controlling TEAD4 isoforms in addition to splicing regulation at RNA level.”

Minor Points

5. Supplementary Figure 2a is not referenced in the Results section.

We have now included the reference of Supplementary Fig. 2a in the results section (page 7, in the main text).

6. The authors should specify more clearly in the Results section and relevant Figure legends when blotting for TEAD4 whether FLAG-tagged or endogenous protein has been assessed, e.g., Figure 1b.

We have included more information in the legend of Fig. 1b.

7. A brief mention of TAZ, a paralog of YAP, and its potentially different mode of interaction with TEAD is suggested in the discussion.

As suggested, we included additional discussion on this topic (page 15, main text).

8. Overall clarity of the manuscript could be improved by further editing for grammar and sentence structure.

We have made further edits in the manuscript to eliminate grammar errors and make it read smoothly.

Reviewer #3 (Remarks to the Author): Expert in Hippo signalling

The manuscript by Qi et al describes a new TEAD4 splicing variant that lacks exon 3. This variant lacks the DNA binding domain and thus cannot promote gene expression but retains the ability interact with YAP. The authors thus propose that this short variant acts as a dominant negative to compete with full-length TEAD4 and inhibit YAP induced gene expression. RBM4 is identified as the splicing factor that promotes exon skipping in TEAD4-S. The authors go on to show that as compared to TEAD4-FL, TEAD-4S is not able to promote cell proliferation, colony formation on soft agar, expression of EMT markers or promote in vivo tumor formation in the context of YAP overexpression. Analysis of cancer databases shows a correlation between TEAD4-S expression levels and overall cancer patient survival at least in lung and colon cancers. The main conclusion drawn by the authors is that the ability of TEAD-S to block YAP function in normal cells is lost due to a decrease in expression of TEAD4-S and this promotes the YAP-induced tumorigenic phenotype. However, while the authors have shown that TEAD-S is incapable of cooperating with overexpressed YAP, insufficient evidence is provided to support the contention the TEAD4-S normally functions to repress YAP activity by competing with endogenous FL TEAD and that loss of TEAD4-S in cancers results in enhanced YAP signalling. Specific comments are provided below.

1. The majority of the experiments compare the effect of overexpressing TEAD FL versus

TEAD S on target gene expression, proliferation etc. However, at best the data shows that TEAD S is not able to promote the observed effect. It does not provide evidence that it can compete with FL TEAD to regulate outcome and thus that a loss would increase YAP signaling. The authors need to focus on the effect of TEAD S as compared to Vector alone

The same point was also raised by reviewer 1. We have now conducted several new experiments to directly confirm that TEAD4-S can indeed compete with TEAD4-FL to regulate YAP-dependent transcriptional activation.

First, we used co-immunoprecipitation assay with anti-Flag antibody to examine if TEAD4-S can directly compete with TEAD4-FL to bind YAP. We co-transfected Flag-TEAD4-FL and HA-YAP with increased amounts of untagged TEAD4-S, and found that TEAD4-S could reduce the co-precipitated HA-YAP in a dose-dependent fashion (new Fig. 2c and page 7 in the main text). This direct competition was further confirmed by precipitating the Flag-YAP in the presence or absence of untagged TEAD4-S, and again we found that TEAD4-S could reduce the interaction between Flag-YAP and endogenous TEAD4-FL in a dose dependent manner (new Supplementary Fig. 2b). These data provided direct evidence to demonstrate that TEAD4-S could directly compete with TEAD4-FL.

Fig. 2c. 293T cells were co-transfected with Flag-TEAD4-FL and HA-YAP, and increasing amounts of untagged TEAD4-S. The interaction between YAP and Flag-TEAD4-FL was determined by co-IP assay.

Supplementary Fig. 2b. 293T cells were co-transfected with Flag-YAP and/or untagged TEAD4-S. The interaction between YAP and endogenous TEAD4-FL was examined by IP assay.

Furthermore, we co-expressed TEAD4-FL with increasing amounts of TEAD4-S, and found a dose dependent reduction of the transcriptional activity of TEAD4-FL as judged by the luciferase assay. We have now modified the Fig. 2a to include these data (new Figure 2a and page 7 in the main text, also see below).

Figure 2a. A luciferase reporter driven by CTGF promoter was co-transfected in the presence or absence of YAP as indicated with TEAD4-FL, TEAD4-S, or RBM4. The relative luciferase activities were determined by calculating the ratio of firefly luciferase activities over Renilla luciferase activities. The data showed TEAD4-S can directly inhibit the transcriptional activity of TEAD4-FL in a dose dependent manner.

Finally, we also specifically switch TEAD4 splicing towards the short isoform with engineered splicing factor (ESF-TEAD4). ESFs are novel tools developed by our lab to manipulate RNA alternative splicing (Wang Y. et. al. Nature Methods 2009). Such splicing switch to TEAD4-S indeed reversed the EMT phenotypes and inhibited tumor cell proliferation as judged by colony formation assay. These results present stronger evidences than overexpressing TEAD-S alone, and were included as a new figure 4 (page 10-11 in the main text).

Figure 4. The splicing switch to TEAD4-S inhibits EMT and cancer cell proliferation. (a) Schematics of the ESF that regulates the splicing of TEAD4. (b) ESF-TEAD4 specifically modulates the splicing of TEAD4 towards the short form in two lung cancer cell lines. The splicing switch to TEAD4-S inhibits EMT (c), and cancer cell proliferation (d) in two different lung cancer cell lines.

2. Given the importance to the study, the authors must describe and show data confirming that the TEAD primers they are using to detect full length and short TEAD are specifically detecting the expected products. There are many ways this could be done (siRNAs, re-PCR of purified DNA with nested primers, cloning and sequencing the PCR product, etc). Neither positive or negative controls are shown in any RT-PCR experiments. It would also be useful if the authors could explicitly state what region the TEAD antibody they are using was raised against and/or recognizes in TEAD.

We have confirmed that the TEAD primers indeed detect expected products by direct sequencing of the gel-purified RT-PCR products (supplementary fig. 1a-1b, see below). In addition, the TEAD4 antibody recognizes amino acids 151-261 of Human TEAD4, which can be used to detect both TEAD4-FL and TEAD4-S. We also included this information in the page 5.

Supplementary Fig. 1a. The two isoforms of TEAD4 were detected by RT-PCR (lane 1), and the products were further confirmed by gel purification and re-amplification (lanes 2 and 3) followed by sanger sequencing.

Supplementary Fig. 1b. The sequences of the PCR products of TEAD4. The yellow sequence is part of exon 2, the blue sequence is the entire sequence of exon 3, and the red sequence is part of exon 4. The upper band of the PCR products includes the yellow, blue, and red sequences, which indicates TEAD4-FL. The lower band only includes the yellow and red sequences, which indicates TEAD4-S.

3. Figure 1c. What is the tag on the transfected YAP? What antibody was used to detect YAP?

We used HA-tagged YAP for transfection, and used anti-HA antibody in the blots. This information was included in figure 1c and its legend for clarity.

4. In most experiments where RNA and protein levels of TEAD4-FL versus -S are compared, (ie Fig. 1d,) the relative ratio of TEAD FL to S mRNA does not correlate with that observed in the protein blotting. Is there an explanation?

In this regard, in Figure 3a, the differences in TEAD4-FL and TEAD4-S expression as assessed by RT-PCR between normal cells and cancer cells is not very convincing, although there is a clear difference in protein levels. The same is true in Figure 4c and d.

We have noticed this disparity between mRNA and protein levels for TEAD4-FL/-S (Figures 3a, and 4c/d), which is also raised by the second reviewer (point 4). This is probably due to differential regulation in translation and/or protein degradation, which is discussed in page 15: *“We also noticed that the mRNA and protein levels of two TEAD4 isoforms are not always consistent across different cell lines and tissues (Fig. 3a and Fig. 5c and 5d), suggesting that the two isoforms may be differentially controlled in the levels of protein translation and/or degradation. This observation adds additional layer of complexity in controlling TEAD4 isoforms in addition to splicing regulation at RNA level.”*

5. Figure 1f-h. What exactly is mut2? Does this sequence comply with a known RBM consensus site?

We have now included the sequence of mut 2 (GTAACG) in page 6, which is another known RBM4 binding site that regulates splicing (Wang Y et al 2012, NSMB, Wang Y et al 2014 Cancer Cell). We used this as a positive control for splicing regulation by RBM4.

6. Figure 2a. For the reporter assay data the authors need to show that the levels of the FL vs S TEAD protein is matched, preferably in an aliquot of the sample subjected to reporter

assay analysis.

We have included the levels of TEAD-S vs TEAD4-FL in supplementary figure 2a, which were from the aliquots of the same set of samples (see below).

Supplementary Fig. 2a. The expression levels of TEAD4-FL, TEAD4-S, and YAP were examined using the aliquots of the same set of samples for luciferase assay.

7. Figure 2b. The experimental design is unclear. Is it an untagged TEAD-S used in lane 4? In this case, the authors should blot using the TEAD antibody to visualize total protein levels, not just with the Flag antibody that only detects FL. In this competition experiment it is important that the ratio of FL to S is assessed. Did the authors try a titration to support the notion that it is a competition? Did the authors try IP'ing YAP and blotting for the FL and S TEAD as an alternative to show how effective the S form is in competing for binding?

Yes, it is an untagged TEAD-S. As suggested, we performed titration experiments to confirm the direct competition (Fig. 2c, also see our response to the first comment of this reviewer).

Briefly, we co-transfected Flag-TEAD4-FL and HA-YAP with increased amounts of untagged TEAD4-S, and found that TEAD4-S could reduce the co-precipitated HA-YAP in a dose-dependent fashion (new Fig. 2c and page 7 in the main text). Also as suggested, this direct competition was further confirmed by precipitating the Flag-YAP with or without untagged TEAD4-S, and again we found that TEAD4-S reduces the interaction between YAP and endogenous TEAD4-FL in a dose dependent manner (Supplementary Fig. 2b).

Fig. 2c. 293T cells were co-transfected with Flag-TEAD4-FL and HA-YAP, and increasing amounts of untagged TEAD4-S. The interaction between YAP and Flag-TEAD4-FL was determined by co-IP.

Supplementary Fig. 2b. 293T cells were co-transfected with Flag-YAP and/or untagged TEAD4-S. The interaction between YAP and endogenous TEAD4-FL was examined by co-IP.

8. Figure 2c-e. The total level of expression of Tead FL vs S and YAP in the stable cell lines must be shown using lysates from cells grown in parallel. In the case of RBM cells, the increase in TEAD S protein should also be shown. A positive control in the case of TEAD S and YAP/RBM samples would also be useful. Do the authors have an explanation for the modest effect of YAP/RBM on Itgb in A549 cells?

As suggested, we have included the expression levels of TEAD4-FL, TEAD4-S, and YAP in the stable cell lines, as well as the TEAD4-S protein level in RBM4 over-expressing cells (new Supplementary Fig. 2c and 2d, page 7-8 in the main text, see below). Although the effect of YAP/RBM4 on *Itgb* in A549 cells was modest, the change was significant as compared to YAP ($p=0.03$, paired T-test).

Supplementary Fig. 2c and 2d. The aliquots of the same set of samples for ChIP (2c), and realtime RT-PCR assays (2d) were used to examine the protein levels of TEAD4, YAP, and RBM4 using western blots.

9. In regards to the RNA-seq data shown in Figure 2f; YAP is thought to function primarily through cooperation with TEAD, so it is unclear why overexpression of YAP + TEAD should give such a dramatically different set of induced genes.

The majority of YAP+TEAD is indeed more overlapped with YAP alone. To reduce the noise and give a better comparison of two TEAD4 isoforms. we identified and only showed the subset of 429 genes that are more sensitive to the differential regulation by TEAD4. See figure below if all genes are included.

Rebuttal Fig. 2 Heat map presentation of the relative expression for all genes in cells expressing YAP (column 1), YAP/TEAD4-FL (column 2), YAP/TEAD4-S (column 3) and YAP/RBM4 (column 4). The genes are ranked according to the levels in YAP transfected cells.

If TEAD-S competes for binding, I would expect that this construct would be more effective against the endogenous TEAD. Have the authors done a detailed analysis of genes downregulated by TEAD-S as compared to empty vector cells or cells expressing YAP alone? Since the authors are arguing the TEAD S can compete with TEAD FL, this would be the most relevant comparison.

This is a good point, however YAP activity is low in the normal growth condition of the cell line used, and thus expressing TEAD4 alone will have small changes on gene expression. Therefore we use the cell line with YAP over-expression to increase the sensitivity. Co-expressing YAP and TEAD4 is a pretty common practice when studying the targets of YAP signaling (for example, Zhao B, Genes Dev, 2008).

Finally, insufficient information is provided to assess the results of the RNAseq. How many genes were increased/decreased in each cell line type (ieYAP, YAP+TEAD-FL, etc) as compared to the vector and/or YAP alone? Are there many genes that are increased in both TEAD FL versus S?

We have included a data table for all the genes that differentially expressed in each condition (Supplementary Table 1). Like most transcriptome-wide profiling, there are some genes that increased in both TEAD FL and S, but their number is small and did not form a large cluster in hierarchical clustering. Therefore we focused on more interested genes.

10. Figure 3e. Quantitation of localization is essential. The authors should also confirm antibody specificity by showing images of YAP staining in non-transfected cells.

As suggested, we included the quantification of localization. All the captured cells that were transfected with both GFP-TEAD4-FL and Flag-YAP have TEAD4-FL localized in the nucleus (~100%, >50 cells in all pictures); While all the cells transfected with GFP-TEAD4-S and YAP have TEAD4-S in both cytoplasm and nucleus (~100%, >50 cells in all pictures).

To further confirm this, we co-transfected Flag-tagged TEAD4-FL or TEAD4-S with myc-YAP, and found the same results (new Supplementary Fig. 3b, page 10, also see below). In cells without TEAD4 transfection, YAP is found in both nucleus and cytoplasm, which is consistent with other reports.

Supplementary Fig. 3b. The cells were co-transfected with myc-YAP and Flag-TEAD4-FL or Flag-TEAD4-S. Then the cells were stained with anti-Flag and anti-myc antibodies and the localizations of YAP (red) and TEAD4 (green) were observed under an immunofluorescence microscope.

11. Figure 4d. Three of the tumor samples show a significant difference in the ratio of S to FL, but in all of the others, the tumor sample is underloaded and thus does not support the authors conclusion that there is a change in the levels. The text should be modified accordingly.

This is indeed due to the heterogeneity of TEAD4 expression level in different tumor samples, and we modified the text according the reviewer's suggestion to reflect the obvious heterogeneity in different tumor specimens (page 12).

12. Supplementary Figure 1a, b. The legends are so brief that it is not possible to easily understand what is being shown. For example, in a, the legend simply says 'mRNA levels'. I suspect this must be a DNA stained gel from an RT-PCR reaction.

The reviewer is correct that the mRNA level is a DNA stained gel from RT-PCR. We have included additional information in the supplementary figure legends to clarify these experimental details. The new legends are: "(c-d) The expression levels of two TEAD4 splicing isoforms in different human tissues. A panel of human tissue cDNAs were used as templates to amplify TEAD4 isoforms, and the PCR products were run on a 10% TBE-PAGE gel to separate. The representative gel figure was shown in (c). A panel of human tissue proteins were applied for a western blot assay. The two isoforms of TEAD4 were blotted with anti-TEAD4 antibody as shown in (d).".

13. Suppl. Fig. 2c is not explained in the results, nor does the legend provide enough detail to understand what point is being made.

The Supplementary Fig. 2c was cited in the main text (page 8, now with new label as supplementary fig. 2g). As suggested, we also included addition information in the legends of this figure.

14. Suppl. Fig. 3. Blots from two cells lines are shown, but the lanes are not labeled to indicate which cell line is which. The tubulin levels for the vector lane for cell line shown on the right are very low, thus the blotting for protein levels, particularly YAP are not convincing.

We have repeated this experiment with even loadings, and included new labels for this figures (Supplementary Fig. 3a, see below).

Supplementary Fig. 3a. H157 and A549 cells that stably expressing YAP alone or co-expressing YAP with TEAD4-FL, TEAD4-s or RBM4 were collected. The protein expression levels of YAP, TEAD4-FL, TEAD4-S, and RBM4 were determined with western blot using antibodies against TEAD4, YAP, or RBM4.

Reviewer #1 had no further comments for the authors.

Reviewer #2 (Remarks to the Author):

This is a very carefully revised manuscript. All points of my critique were addressed well.

Reviewer #3 (Remarks to the Author):

The authors have adequately address my comments. However, there remain numerous grammatical errors that significantly detract from the presentation of the results. As an example, every sentence in the abstract has an error (such as the use of plurals and 'the' included or excluded incorrectly).